# Bayesian Generational Population-Based Training

**Xingchen Wan**[1]  **Cong Lu**[1]  **Jack Parker-Holder**[1]  **Philip J. Ball**[1]  **Vu Nguyen**[2]  **Binxin Ru**[1]
**Michael A. Osborne**[1]

[1]Machine Learning Research Group, University of Oxford, Oxford, UK
[2]Amazon, Adelaide, Australia

**Abstract**  Reinforcement learning (RL) offers the potential for training generally capable agents that can interact autonomously in the real world. However, one key limitation is the brittleness of RL algorithms to core hyperparameters and network architecture choice. Furthermore, non-stationarities such as evolving training data and increased agent complexity mean that different hyperparameters and architectures may be optimal at different points of training. This motivates AutoRL, a class of methods seeking to automate these design choices. One prominent class of AutoRL methods is Population-Based Training (PBT), which have led to impressive performance in several large scale settings. In this paper, we introduce two new innovations in PBT-style methods. First, we employ trust-region based Bayesian Optimization, enabling full coverage of the high-dimensional mixed hyperparameter search space. Second, we show that using a *generational* approach, we can also learn both architectures and hyperparameters jointly on-the-fly in a single training run. Leveraging the new highly parallelizable Brax physics engine, we show that these innovations lead to large performance gains, significantly outperforming the tuned baseline while learning entire configurations on the fly. Code is available at `https://github.com/xingchenwan/bgpbt`.

## 1 Introduction

Reinforcement Learning (RL) (Sutton and Barto, 2018) has proven to be a successful paradigm for training agents across a variety of domains and tasks (Kalashnikov et al., 2018; Mnih et al., 2013; Nguyen et al., 2021b; Silver et al., 2017), with some believing it could be enough for training generally capable agents (Silver et al., 2021). However, a crucial factor limiting the wider applicability of RL to new problems is the notorious sensitivity of algorithms with respect to their hyperparameters (Andrychowicz et al., 2021; Engstrom et al., 2020; Henderson et al., 2018), which often require expensive tuning. Indeed, it has been shown that when tuned effectively, good configurations often lead to dramatically improved performance in large scale settings (Chen et al., 2018).

To address these challenges, recent work in *Automated Reinforcement Learning* (AutoRL) (Parker-Holder et al., 2022) has shown that rigorously searching these parameter spaces can lead to previously unseen levels of performance, even capable of breaking widely used simulators (Zhang et al., 2021). However, AutoRL contains unique challenges, as different tasks even in the same suite are often best solved with different network architectures and hyperparameters (Furuta et al., 2021; Xu et al., 2022). Furthermore, due to the non-stationarities present in RL (Igl et al., 2021), such as changing data distributions and the requirement for agents to model increasingly complex behaviors over time, optimal hyperparameters and architectures may not remain constant. To address this, works have shown adapting hyperparameters through time (Jaderberg et al., 2017; Parker-Holder et al., 2021; Paul et al., 2019; Zhang et al., 2021) and defining fixed network architecture schedules (Czarnecki et al., 2018) can be beneficial for performance. However, architectures and hyperparameters are inherently linked (Park et al., 2019), and to date, no method combines the ability to jointly and continuously adapt both on the fly.

In this paper we focus on Population-based Training (PBT) (Jaderberg et al., 2017) methods, where a population of agents is trained in parallel, copying across stronger weights and enabling

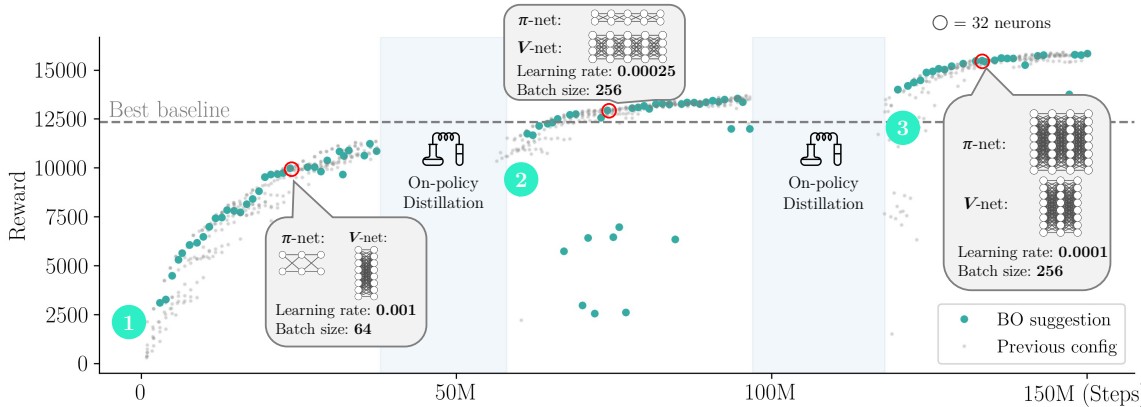

Figure 1: An example run of BG-PBT on HalfCheetah task in BRAX: BG-PBT combines population-based training with high-dimensional Bayesian optimization, generational training (different generations marked with numbers in the figure) and on-policy distillation between generations to transfer across RL agents with different neural architectures: at different points during training, both hyperparameters and the architectures of policy & value networks are tuned on-the-fly, leading to significant improvement over the baseline.

adaption of hyperparameters in a single training run. This allows PBT methods to achieve impressive performance on many large-scale settings (Jaderberg et al., 2019; Liu et al., 2021). However, PBT-style methods are typically limited in scope due to two key factors: 1) they only optimize a handful of hyperparameters, either due to using random search (Jaderberg et al., 2017), or model-based methods that do not scale to higher dimensions (Parker-Holder et al., 2021, 2020); 2) PBT methods are usually restricted to the same fixed architecture since weights are copied between agents.

We seek to overcome both of these issues in this paper, and propose *Bayesian Generational Population-based Training (BG-PBT)*, with an example run demonstrated in Fig. 1. BG-PBT is capable of tuning a significantly greater proportion of the agent's configuration, thanks to two new ideas. First, we introduce a new model-based *hyperparameter and architecture* exploration step motivated by recent advances in local Bayesian optimization (Wan et al., 2021). Second, we take inspiration from Stooke et al. (2021) who showed that PBT can be particularly effective when combined with network distillation (Igl et al., 2021), in an approach known as generational learning. As prior works in generational training (Stooke et al., 2021; Vinyals et al., 2019) show, the use of successive generations of architectures with distillation results in significantly reduced training time for new agents. This provides us with an algorithm-agnostic framework to create agents which continuously discover their *entire configuration*. Thus, for the first time, we can tune hyperparameters and architectures during one training run as part of a single unified algorithm.

We run a series of exhaustive experiments tuning both the architectures and hyperparameters for a Proximal Policy Optimization (PPO) (Schulman et al., 2017) agent in the newly introduced BRAX environment suite (Freeman et al., 2021). BRAX enables massively parallel simulation of agents, making it perfect for testing population-based methods without vast computational resources. Our agents significantly outperform both the tuned baseline and a series of prior PBT methods. Notably, we observe that BG-PBT often discovers a *schedule of networks* during training—which would be infeasible to train from scratch. Furthermore, BG-PBT discovers entirely new modes of behavior for these representative environments, which we show at https://sites.google.com/view/bgpbt.

To summarize, the main contributions of this paper are as follows:

1. We show for the first time it is possible to select architectures as part of a general-purpose PBT framework, using *generational training with policy distillation* with Neural Architecture Search (NAS).

2. We propose a novel and efficient algorithm, BG-PBT, especially designed for high-dimensional mixed search spaces, which can select both architectures and hyperparameters on-the-fly with provable efficiency guarantees.
3. We show in a series of experiments our *automatic architecture curricula* make it possible to achieve significantly higher performance than previous methods.

## 2 Preliminaries

We begin by introducing the reinforcement learning framework, population-based training, which our method is based on, and the general problem setup we investigate in this paper.

**Reinforcement Learning.** We model the environment as a Markov Decision Process (MDP) (Sutton and Barto, 2018), defined as a tuple $M = (\mathcal{S}, \mathcal{A}, P, R, \rho_0, \gamma)$, where $\mathcal{S}$ and $\mathcal{A}$ denote the state and action spaces respectively, $P(s_{t+1}|s_t, a_t)$ the transition dynamics, $R(s_t, a_t)$ the reward function, $\rho_0$ the initial state distribution, and $\gamma \in (0, 1)$ the discount factor. The goal is to optimize a policy $\pi(a_t|s_t)$ that maximizes the expected discounted return $\mathbb{E}_{\pi,P,\rho_0}\left[\sum_{t=0}^{\infty} \gamma^t R(s_t, a_t)\right]$. Given a policy $\pi$, we may define the state value function $V^{\pi}(s) = \mathbb{E}_{\pi,P}\left[\sum_{t=0}^{\infty} \gamma^t R(s_t, a_t)|s_0 = s\right]$ and the state-action value-function $Q^{\pi}(s, a) = \mathbb{E}_{\pi,P}\left[\sum_{t=0}^{\infty} \gamma^t R(s_t, a_t)|s_0 = s, a_0 = a\right]$. The advantage function is then defined as the difference $A^{\pi}(s, a) = Q^{\pi}(s, a) - V^{\pi}(s)$.

A popular algorithm for online continuous control that we use is PPO (Schulman et al., 2017). PPO achieves state-of-the-art results for popular benchmarks (Cobbe et al., 2020) and is hugely parallelizable, making it an ideal candidate for population-based methods (Parker-Holder et al., 2020). PPO approximates TRPO (Schulman et al., 2015, 2017) and uses a clipped objective to stabilize training:

$$\mathcal{L}_{\text{PPO}}(\theta) = \min\left(\frac{\pi_{\theta}(a \mid s)}{\pi_{\mu}(a \mid s)} A^{\pi_{\mu}}, g(\theta, \mu) A^{\pi_{\mu}}\right), \text{ where } g(\theta, \mu) = \text{clip}\left(\frac{\pi_{\theta}(a \mid s)}{\pi_{\mu}(a \mid s)}, 1 - \epsilon, 1 + \epsilon\right) \quad (1)$$

where $\pi_{\mu}$ is a previous policy and $\epsilon$ is the clipping parameter.

**Population-Based Training.** RL algorithms, including PPO, are typically quite sensitive to their hyperparameters. PBT (Jaderberg et al., 2017) is an evolutionary method that tunes RL hyperparameters on-the-fly. It optimizes a population of $B$ agents in parallel, so that their weights and hyperparameters may be dynamically adapted within a single training run. In the standard paradigm without architecture search, we consider two sub-routines, explore and exploit. We train for a total of $T$ steps and evaluate performance every $t_{\text{ready}} < T$ steps. In the exploit step, the weights of the worst-performing agents are replaced by those from an agent randomly sampled from the set of best-performing ones, via *truncation selection*. To select new hyperparameters, we perform the explore step. We denote the hyperparameters for the $b$th agent in a population at timestep $t$ as $\mathbf{z}_t^b \in \mathcal{Z}$; this defines a *schedule* of hyperparameters over time $(z_t^b)_{t=1,\dots T}$. Let $f_t(z_t)$ be an objective function (e.g. the return of a RL agent) under a given set of hyperparameters at timestep $t$, our goal is to maximize the final performance $f_T(z_T)$.

The original PBT uses a combination of random sampling and evolutionary search for the explore step by suggesting new hyperparameters mutated from the best-performing agents. Population Based Bandit (PB2) and PB2-Mix (Parker-Holder et al., 2021, 2020) improve on PBT by using *Bayesian optimization (BO)* to suggest new hyperparameters, relying on a time-varying *Gaussian Process (GP)* (Bogunovic et al., 2016; Rasmussen and Williams, 2006) to model the data observed. We will also use GP-based BO in our method, and we include a primer of GPs and BO in App. A.

**Problem Setup.** We follow the notation used in Parker-Holder et al. (2020) and frame the hyperparameter optimization problem in the lens of optimizing an expensive, time-varying, black-box reward function $f_t : \mathcal{Z} \to \mathbb{R}$. Every $t_{\text{ready}}$ steps, we observe and record noisy observations, $y_t = f_t(\mathbf{z}_t) + \epsilon_t$, where $\epsilon_t \sim \mathcal{N}(0, \sigma^2\mathbf{I})$ for some fixed variance $\sigma^2$. We follow the typical PBT setup by defining a hyperparameter space, $\mathcal{Z}$, which for the BRAX (Freeman et al., 2021) implementation

of PPO we follow in the paper, consists of 9 parameters: learning rate, discount factor ($\gamma$), entropy coefficient ($c$), unroll length, reward scaling, batch size, updates per epoch, GAE parameter ($\lambda$) and clipping parameter ($\epsilon$). To incorporate the architecture hyperparameters, $\mathbf{y} \in \mathcal{Y}$, we add 6 additional parameters leading to a 15-dimensional joint space $\mathcal{J} = \mathcal{Y} \times \mathcal{Z}$. For both the policy and value networks, we add the width and depth of the Multi-layer Perceptron (MLP) and a binary flag on whether to use spectral normalization.

## 3 Bayesian Generational Population-Based Training (BG-PBT)

We present BG-PBT in Algorithm 1 which consists of two major components. First, a BO approach to select new hyperparameter configurations $\mathbf{z}$ for our agents (§3.1). We then extend the search space to accommodate architecture search, allowing agents to choose their own networks (parameterized by $\mathbf{y} \in \mathcal{Y}$) and use on-policy distillation to transfer between different architectures (§3.2).

### 3.1 High-Dimensional BO Agents in Mixed-Input Configuration Space for PBT

Existing population-based methods ignore (PB2) or only partially address (PB2-Mix, which does not consider ordinal variables such as integers) the heterogeneous nature of the mixed hyperparameter space $\mathcal{Z}$. Furthermore, both previous methods are equipped with standard GP surrogates which typically scale poorly beyond low-dimensional search spaces, and are thus only used to tune a few selected hyperparameters deemed to be the most important based on human expertise. To address these issues, BG-PBT

---

**Algorithm 1** BG-PBT; distillation and NAS steps marked in magenta (§3.2)

1: **Input**: pop size $B$, $t_{\text{ready}}$, max steps $T$, $q$ (% agents replaced per iteration)
2: **Initialize** $B$ agents with weights $\{\theta_0^{(i)}\}_{i=1}^B$, random hyperparameters $\{\mathbf{z}_0^{(i)}\}_{i=1}^B$ and architectures $\{\mathbf{y}_0^{(i)}\}_{i=1}^B$,
3: **for** $t = 1, \ldots, T$ (in parallel for all $B$ agents) **do**
4:     Train models & record data for all agents
5:     **if** $t \bmod t_{\text{ready}} = 0$ **then**
6:         Replace the weights & architectures of the bottom $q$% agents with those of the top $q$% agents.
7:         Update the surrogate with new observations & returns and adjust/restart the trust regions.
8:         Check whether to start a new generation (see §3.2).
9:         **if** start a new generation **then**
10:         Clear the GP training data.
11:         Create $B$ agents with archs. from BO/random.
12:         Distill from a top-$q$% performing agent of the existing generation to new agents.
13:         **else**
14:         Select new hyperparameters $\mathbf{z}$ for the agents whose weights have been just replaced with randomly sampled configs (if $\mathbf{D} = \emptyset$) **OR** using the suggestions from the BO agent described conditioned on $\mathbf{y}$ (otherwise).

---

explicitly accounts for the characteristics of typical RL hyperparameter search space by making several novel extensions to CASMOPOLITAN (Wan et al., 2021), a state-of-the-art BO method for high-dimensional, mixed-input problems for our setting. In this section, we outline the main elements of our design, and we refer the reader to App. B.2 for full technical details of the approach.

**Tailored Treatment of Mixed Hyperparameter Types.** Hyperparameters in RL can be continuous (e.g. discounting factor), ordinal (discrete variables with ordering, e.g. batch size) and categorical (discrete variables without ordering, e.g. activation function). BG-PBT treats each variable type differently: we use tailored kernels for the GP surrogate, and utilize interleaved optimization for the acquisition function, alternating between local search for the categorical/ordinal variables and gradient descent for the continuous variables. BG-PBT extends both CASMOPOLITAN and PB2-Mix by further accommodating ordinal variables, as both previous works only considered continuous and categorical variables. We demonstrate the considerable benefits of explicitly accounting for the ordinal variables in App. F.6.

**Trust Regions (TR).** TRs have proven success in extending GP-based BO to higher-dimensional search spaces, which were previously intractable due to the curse of dimensionality, by limiting exploration to promising regions in the search space based on past observations (Eriksson et al., 2019; Wan et al., 2021). In the PBT context, TRs also implicitly avoid large jumps in hyperparameters, which improves training stability. We adapt the TRs used in the original CASMOPOLITAN to the time-varying setup by defining TRs around the current best configuration, and then adjusting them

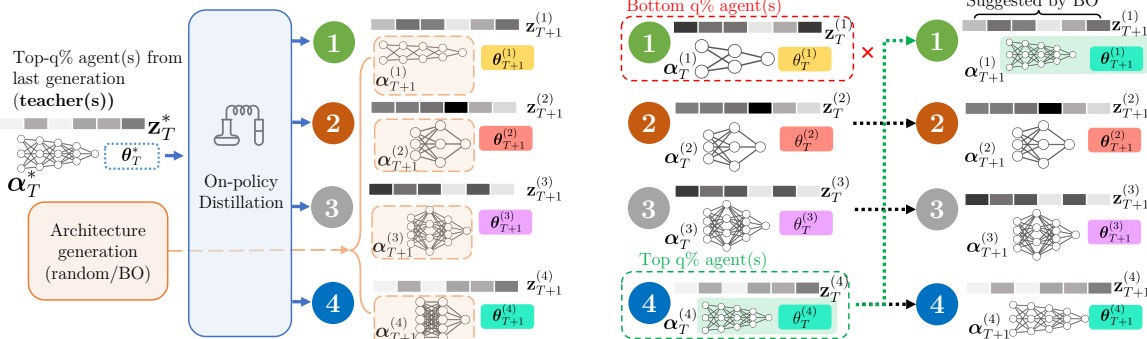

Figure 2: BG-PBT (a) at the *beginning* of a generation (**left**) and (b) *during* a generation (**right**). At the start of a generation, agents with diverse architectures are suggested and on-policy distillation is used to transfer information across generations & different architectures (§3.2). Within a generation, a high-dimensional, mixed-input BO agent suggests hyperparameters (§3.1, we copy weights across fixed architectures).

dynamically: similar to Eriksson et al. (2019) and Wan et al. (2021), TRs are expanded or shrunk upon consecutive "successes" or "failures". We define a proposed configuration to be a "success" if it appears in the top $q$%-performing agents and a "failure" otherwise. When the TRs shrink below some specified minimum, a *restart* is triggered, which resets the GP surrogate to avoid becoming stuck at a local optimum. We adapt the Upper Confidence Bound (UCB)-based criterion proposed in Wan et al. (2021) which is based on a global, auxiliary GP model to the time-varying setting to re-initialize the population when a restart is triggered. Full details are provided in App. B.4.

**Theoretical Properties.** Following Wan et al. (2021), we show that under typical assumptions (presented in App. C) used for TR-based algorithms (Yuan, 1999), our proposed BG-PBT, without architecture and distillation to be introduced in §3.2, converges to the global optimum asymptotically. Furthermore, we derive an upper bound on the cumulative regret and show that under certain conditions it achieves sublinear regret. We split the search space into $\mathcal{Z} = [\mathcal{H}, \mathcal{X}]$ (categorical/continuous parts respectively). We note that Assumption C.3 considers the minimum TR lengths $L_{\min}^x, L_{\min}^h$ are set to be small enough so that the GP approximates $f$ accurately in the TRs. In practice, this assumption only holds asymptotically, i.e. when the observed datapoints in the TRs goes to infinity. We present the main result, the time-varying extension to Theorem 3.4 from Wan et al. (2021), and then refer to App. C for the derivation.

**Theorem 3.1.** *Assume Assumptions C.2 & C.3 hold. Let $f_i : [\mathcal{H}, \mathcal{X}] \to \mathbb{R}$ be a time-varying objective defined over a mixed space and $\zeta \in (0, 1)$. Suppose that: (i) there exists a class of functions $g_i$ in the RKHS $\mathcal{G}_k([\mathcal{H}, \mathcal{X}])$ corresponding to the kernel $k$ of the global GP model, such that $g_i$ passes through all the local maximas of $f_i$ and shares the same global maximum as $f_i$; (ii) the noise at each timestep $\epsilon_i$ has mean zero conditioned on the history and is bounded by $\sigma$; (iii) $\|g_i\|_k^2 \leq B$. Then BG-PBT obtains a regret bound*

$$Pr\left\{ R_{IB} \leq \sqrt{\frac{C_1 I \beta_I}{B} \gamma\big(IB; k; [\mathcal{H}, \mathcal{X}]\big)} + 2 \quad \forall I \geq 1 \right\} \geq 1 - \zeta,$$

*with $C_1 = 8/\log(1 + \sigma^{-2})$, $\gamma(T; k; [\mathcal{H}, \mathcal{X}])$ defined in Theorem C.1 and $\beta_I$ is parameter balancing exploration-exploitation as in Theorem 2 of Parker-Holder et al. (2020).*

Under the same ideal conditions assumed in Bogunovic et al. (2016); Parker-Holder et al. (2020) where the objective does not vary significantly through time, the cumulative regret bound is sublinear with $\lim_{I \to \infty} \frac{R_{IB}}{I} = 0$, when $\omega \to 0$ and $\tilde{N} \to I$.

## 3.2 Adapting Architectures on the Fly

Now that we are equipped with an approach to optimize in high-dimensional $\mathcal{Z}$, we focus on choosing *network architectures*. Despite their importance in RL (Cobbe et al., 2019; Furuta et al., 2021), architectures remain underexplored as a research direction. Adapting architectures for PBT methods is non-trivial as we further enlarge the search space, and weights cannot readily be copied across different networks. Inspired by Stooke et al. (2021), our key idea is that when beginning a new *generation* we can distill behaviors into new architectures (see Fig. 2). Specifically:

- *Starting each generation:* We fill the population of $B$ agents by generating a diverse set of architectures for both the policy and value networks. For the first generation, this is done via random sampling. For subsequent generations, we use suggestions from BO and/or random search with successive halving over the architecture space $\mathcal{Y}$ only (refer to App. B.3 for details); the BO is trained on observations of the best performance each architecture has achieved in previous generations. We initialize a new generation when the evaluated return stagnates beyond a pre-set patience during training.

- *Transfer between generations:* Apart from the very first generation, we transfer information from the best agent(s) of the previous generation to each new agent, in a similar fashion to Stooke et al. (2021), using on-policy distillation with a joint supervised and RL loss between *different architectures* as shown in Fig. 2a. Given a learned policy $\pi_i$ and value function $V_i$ from a previous generation, the new joint loss optimized is:

$$\mathbb{E}_{(s_t, a_t) \sim \pi_{i+1}} \left[ \alpha_{RL} \mathcal{L}_{RL} + \alpha_V \left\| V_i(s_t) - V_{i+1}(s_t) \right\|_2 + \alpha_\pi \mathbb{D}_{KL} \left( \pi_i(\cdot \mid s_t) \mid\mid \pi_{i+1}(\cdot \mid s_t) \right) \right] \quad (2)$$

for weights $\alpha_{RL} \geq 0$, $\alpha_V \geq 0$, $\alpha_\pi \geq 0$, and RL loss $\mathcal{L}_{RL}$ taken from Equation (1). We linearly anneal the supervised losses over the course of each generation, so that by the end, only the RL loss remains.

- *During a generation:* We follow standard PBT methods to evolve the hyperparameters of each agent by copying weights $\theta$ *and the architecture* y from a top-$q$% performing agent to a bottom-$q$% agent, as shown in Fig. 2b. This creates an effect similar to successive halving (Jamieson and Talwalkar, 2016; Karnin et al., 2013) where poorly-performing architectures are quickly removed from the population in favor of more strongly-performing ones; typically at the end of a generation, 1 or 2 architectures dominate the population. While we do not introduce new architectures within a generation, the hyperparameter suggestions are conditioned on the current policy and value architectures by incorporating the architecture parameters y as contextual fixed dimensions in the GP surrogate described in §3.1.

## 4 Experiments

While BG-PBT provides a framework applicable to any RL algorithm, we test our method on 7 environments from the new Brax environment suite, using PPO. We begin by presenting a comparative evaluation of BG-PBT against standard baselines in population-based training to both show the benefit of searching over the full hyperparameter space with local BO and of automatically adapting architectures over time. We further show that our method beats end-to-end BO, showing the advantage of dynamic schedules. Next, we analyze these learned hyperparameter and architecture schedules using BG-PBT and we show analogies to similar trends in learning rate and batch size in supervised learning. Finally, we perform ablations on individual components of BG-PBT. For all population-based methods, we use a population size $B = 8$ and a total budget of 150M steps. We note that BG-PBT with architectures uses additional on-policy samples from the environment in order to distill between architectures. We instantiate the Brax environments with an action repeat of 1. We use $t_{ready}$ of 1M for all PBT-based methods on all environments except for Humanoid and Hopper, where we linearly anneal $t_{ready}$ from 5M to 1M. The remaining hyperparameters and implementation details used in this section are listed in App. E.

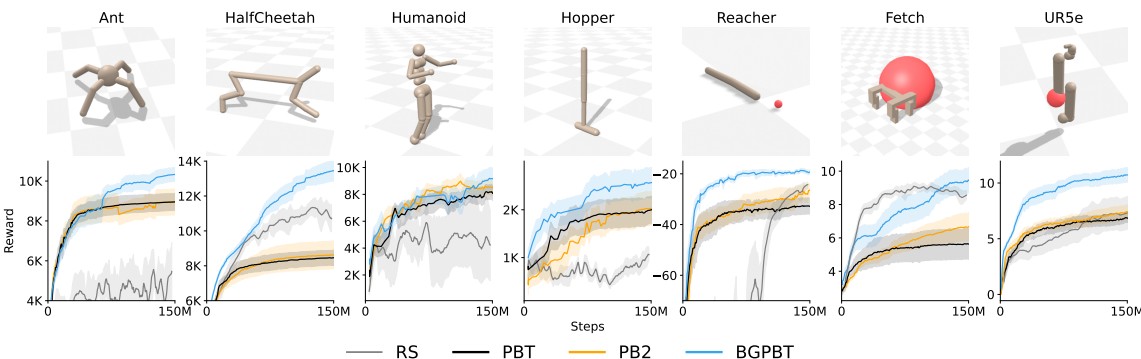

Figure 3: Visualization of each environment (**top row**) and mean evaluated return over the population with ±1 SEM (shaded) across 7 random seeds (**bottom row**) in all environments. RS refers to the higher performing of RS-$\mathcal{Z}$ or RS-$\mathcal{J}$ in Table 1.

Table 1: Mean evaluated return ±1SEM across 7 seeds shown. For PBT-style methods (PBT, PB2 and BG-PBT), the mean best-performing agent in the population is shown. Methods performing within 1 SEM of the best-performing method are bolded (the same applies to all tables).

| Method | PPO* | RS | RS | PBT | PB2 | BG-PBT |
| Search space | $\mathcal{Z}$ | $\mathcal{Z}$ | $\mathcal{J}$ | $\mathcal{Z}$ | $\mathcal{Z}$ | $\mathcal{J}$ |
|---|---|---|---|---|---|---|
| Ant | $3853_{\pm676}$ | $6780_{\pm317}$ | $4781_{\pm515}$ | $8955_{\pm385}$ | $8954_{\pm594}$ | $\mathbf{10349_{\pm326}}$ |
| HalfCheetah | $6037_{\pm236}$ | $9502_{\pm76}$ | $10340_{\pm329}$ | $8455_{\pm400}$ | $8629_{\pm746}$ | $\mathbf{13450_{\pm551}}$ |
| Humanoid | $\mathbf{9109_{\pm987}}$ | $4004_{\pm519}$ | $4652_{\pm1002}$ | $7954_{\pm437}$ | $8452_{\pm512}$ | $\mathbf{9171_{\pm748}}$ |
| Hopper | $120_{\pm43}$ | $339_{\pm25}$ | $943_{\pm185}$ | $2002_{\pm254}$ | $2027_{\pm323}$ | $\mathbf{2569_{\pm293}}$ |
| Reacher | $-189.3_{\pm43.7}$ | $-24.2_{\pm1.4}$ | $-95.2_{\pm25.3}$ | $-32.9_{\pm2.8}$ | $-26.6_{\pm2.6}$ | $\mathbf{-19.2_{\pm0.9}}$ |
| Fetch | $\mathbf{14.0_{\pm0.2}}$ | $5.2_{\pm0.4}$ | $8.6_{\pm0.2}$ | $5.5_{\pm0.8}$ | $6.6_{\pm0.7}$ | $9.4_{\pm0.7}$ |
| UR5e | $5.2_{\pm0.2}$ | $5.3_{\pm0.4}$ | $7.7_{\pm0.3}$ | $6.9_{\pm0.4}$ | $7.4_{\pm0.6}$ | $\mathbf{10.7_{\pm0.6}}$ |

*From the BRAX authors and implemented in a different framework (JAX) to ours (PyTorch)

**Comparative Evaluation of BG-PBT.** We first perform a comparative evaluation of BG-PBT against standard baselines in PBT-methods and the PPO baseline provided by the BRAX authors. We show the benefit of using local BO and treating the whole RL hyperparameter space $\mathcal{Z}$, by comparing BG-PBT against PBT (Jaderberg et al., 2017), PB2 (Parker-Holder et al., 2020) and Random Search (RS) using the default architecture in BRAX. In RS, we simply sample from the hyperparameter space and take the best performance found using the same compute budget as the PBT methods. Next, we include architecture search into BG-PBT using the full space $\mathcal{J}$ and show significant gains in performance compared to BG-PBT without architectures; we use random search over $\mathcal{J}$ as a baseline. The optimized PPO implementation from the BRAX authors is provided as a sequential baseline. We present the results in Table 1 and the training trajectories in Fig. 3.

We show that BG-PBT significantly outperforms the RS baselines and the existing PBT-style methods in almost all environments considered. We also observe that RS is a surprisingly strong baseline, performing on par or better than PBT and PB2 in HalfCheetah, Reacher, Fetch and UR5e — this is due to a well-known failure mode in PBT-style algorithms where they may be overly greedy in copying sub-optimal behaviors early on and then fail to sufficiently explore in weight space when the population size is modest. BG-PBT avoids this problem by *re-initializing networks each generation* and distilling, which prevents collapse to suboptimal points in weight space.

**Experiments with Different Training Timescales**. We further show experiments with a higher budget of 300M timesteps and/or an increased population size up to $B = 24$ to investigate the scalability of BG-PBT in larger-scale environments (App. F.1). Furthermore, given that BG-PBT uses additional on-policy samples in order to distill between architectures, we conduct experiments of BG-PBT with reduced training budget in App. F.2. We find that even when the maximum timesteps are roughly halved, BG-PBT still outperforms the baseline AutoRL methods.

Table 2: Comparison against sequential BO*

| Method | BO-$\mathcal{Z}^*$ | BO-$\mathcal{J}^*$ | BG-PBT |
|---|---|---|---|
| Ant | $6975_{\pm 1013}$ | $7149_{\pm 507}$ | $\mathbf{10349_{\pm 326}}$ |
| HalfCheetah | $11202_{\pm 204}$ | $10859_{\pm 174}$ | $\mathbf{13450_{\pm 551}}$ |
| Humanoid | $9040_{\pm 1303}$ | $4845_{\pm 962}$ | $\mathbf{9171_{\pm 748}}$ |
| Hopper | $358_{\pm 60}$ | $1254_{\pm 154}$ | $\mathbf{2569_{\pm 293}}$ |
| Reacher | $\mathbf{-17.3_{\pm 0.3}}$ | $-51.7_{\pm 18.3}$ | $-19.2_{\pm 0.9}$ |
| Fetch | $\mathbf{13.2_{\pm 0.2}}$ | $11.6_{\pm 0.1}$ | $9.4_{\pm 0.7}$ |
| UR5e | $9.0_{\pm 0.5}$ | $6.3_{\pm 1.4}$ | $\mathbf{10.7_{\pm 0.6}}$ |

*More resources required compared to BG-PBT.

Table 3: Ablation studies

| Method | No TR/NAS | No NAS | BG-PBT |
|---|---|---|---|
| Ant | $8954_{\pm 594}$ | $9352_{\pm 402}$ | $\mathbf{10349_{\pm 326}}$ |
| HalfCheetah | $8629_{\pm 746}$ | $9483_{\pm 626}$ | $\mathbf{13450_{\pm 551}}$ |
| Humanoid | $8452_{\pm 512}$ | $\mathbf{10359_{\pm 647}}$ | $9171_{\pm 748}$ |
| Hopper | $2027_{\pm 323}$ | $2511_{\pm 154}$ | $\mathbf{2569_{\pm 293}}$ |
| Reacher | $-26.6_{\pm 2.6}$ | $\mathbf{-17.6_{\pm 0.8}}$ | $-19.2_{\pm 0.9}$ |
| Fetch | $6.6_{\pm 0.7}$ | $7.3_{\pm 0.8}$ | $\mathbf{9.4_{\pm 0.7}}$ |
| UR5e | $7.4_{\pm 0.6}$ | $9.0_{\pm 0.8}$ | $\mathbf{10.7_{\pm 0.6}}$ |

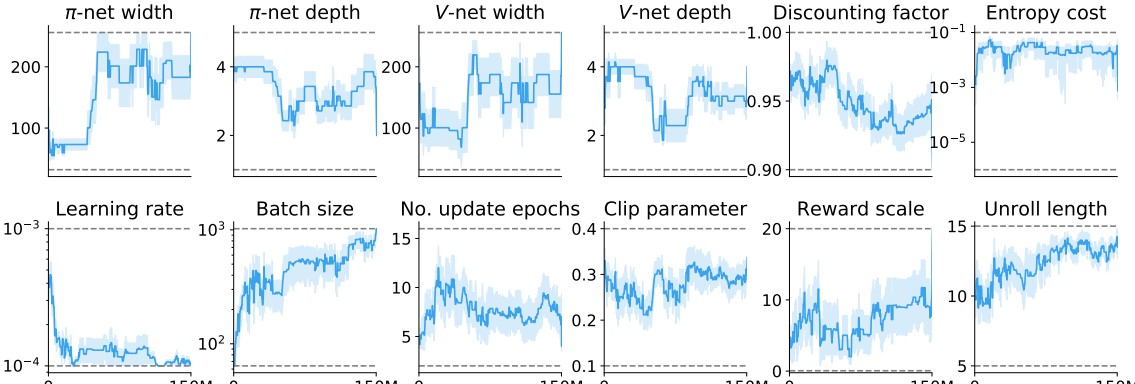

Figure 4: The hyperparameter and architecture schedule discovered by BG-PBT on Ant: we plot the hyperparameters of the best-performing agent in the population averaged across 7 seeds with ± 1 SEM shaded. Gray dashed lines denote the hyperparameter bounds.

**Comparison Against Sequential BO.** We further compare against BO in the traditional sequential setup (Table 2): for each BO iteration, the agent is trained for the full 150M timesteps before a new hyperparameter suggestion is made. To enable BO to improve on RS, we allocate a budget of 50 evaluations, *which is up to 6× more expensive* than our method and even more costly in terms of wall-clock time if vanilla, non-parallel BO is used. We implement this baseline using SMAC3 (Lindauer et al., 2022) in both the $\mathcal{Z}$ and $\mathcal{J}$ search spaces (denoted BO-$\mathcal{Z}$ and BO-$\mathcal{J}$ respectively in Table 2). While, unsurprisingly, BO improves over the RS baseline, BG-PBT still outperforms it in a majority of environments. One reason for this is that BG-PBT naturally discovers a dynamic schedule of hyperparameters and architectures, which is strictly more flexible than a carefully tuned but still static configuration – we analyze this below.

**Analysis of Discovered Hyperparameter and Architecture Schedules.** We present the hyperparameter and architecture schedules learned by BG-PBT in our main comparative evaluation on Ant in Fig. 4 (results on other environments are presented in App. F.4). We find consistent trends across environments such as the decrease of learning rate and increase in batch sizes over time, consistent to common practices in both RL (Engstrom et al., 2020) and supervised learning, but crucially BG-PBT discovers the same *without any pre-defined schedule*. We note, however, that the exact rate at which the learning rate decreases and batch size increase differs across different environments – for example, in Ant we find that the learning rate quickly drops from a relatively large value to almost the smallest possible $10^{-4}$, whereas in UR5e, the schedule is much less aggressive. This suggests that the optimal schedule is dependent on the exact environment, and a uniform, manually-defined schedule as in Engstrom et al. (2020) may not be optimal. We demonstrate this empirically in App. F.7, where we compare against RS but with the learning rate following a manually-defined cosine annealing schedule. We also find that different networks are favored at different stages of training, but the exact patterns differ across environments: for Ant (Fig. 4), we find that larger networks are preferred towards the end of training, with the policy and value network widths increasing over time: Prior work has shown that larger networks like those we automatically find towards the end

of training *can be notoriously unstable and difficult to train from scratch* (Czarnecki et al., 2018; Ota et al., 2021), which further supports our use of generational training to facilitate this.

**Ablation Studies**. BG-PBT improves on existing methods by using local TR-based BO (§3.1) and NAS & distillation (§3.2). We conduct an ablation study by removing either or both components in Table 3 (a comparison between the training trajectories in Fig. 7 may be found in App. F.3), where "No NAS" does not search architectures or distill but uses the default BRAX architectures, and "No TR/NAS" further only uses a vanilla GP surrogate and is identical to PB2. We find the tailored BO agent in §3.1 improves performance across the board. On the importance of NAS & distillation, in all environments except for Humanoid and Reacher, BG-PBT matches or outperforms "No NAS", despite $\mathcal{J}$ being a more complicated search space and the "No NAS" baseline is conditioned on strongly-performing default architectures. We also see a particularly large gain for HalfCheetah and Fetch when we include architectures, demonstrating the effectiveness of the generational training and NAS in our approach. We include additional ablation studies in App. F.3.

## 5 Related Work

**On-the-fly Hyperparameter Tuning**. Our work improves on previous PBT (Jaderberg et al., 2017; Parker-Holder et al., 2020; Zhang et al., 2021) style methods; in particular, we build upon Parker-Holder et al. (2021), using a more scalable BO step, and adding architecture search with generational learning. Dalibard and Jaderberg (2021) introduce an approach for increasing diversity in the weight space for PBT, orthogonal to our work. There have also been non-population-based methods for dynamic hyperparameter optimization, using bandits (Badia et al., 2020; Ball et al., 2020; Moskovitz et al., 2021; Nguyen et al., 2020; Parker-Holder et al., 2020), gradients (Flennerhag et al., 2021; Paul et al., 2019; Xu et al., 2018; Zahavy et al., 2020) or Evolution (Tang and Choromanski, 2020) which mostly do not search over architectures. A notable exception is Sample-efficient Automated Deep Learning (SEARL) (Franke et al., 2021), which adapts architectures within a PBT framework. However, SEARL is designed for off-policy RL and thus especially shows the benefit of shared replay buffers for efficiency, whereas our method is general-purpose.

**Architecture Search**. In RL, Czarnecki et al. (2018) showed increasing agent complexity over time could be effective, albeit with a pre-defined schedule. Miao et al. (2021) showed that DARTS (Liu et al., 2019) could be effective in RL, finding high performing architectures on the Procgen benchmark. Auto-Agent-Distiller (Fu et al., 2020) deals with the problem of finding optimal architectures for compressing the model size of RL agents, and also find that using distillation between the teacher and student networks improves stability of NAS in RL. On the other hand, BO has been used as a powerful tool for searching over large architecture spaces (Kandasamy et al., 2018; Nguyen et al., 2021a; Ru et al., 2021; Wan et al., 2021, 2022; White et al., 2021). Conversely, we only consider simple MLPs and the use of spectral normalization. There has been initial effort (Izquierdo et al., 2021) combining NAS and hyperparameter optimization in sequential settings, which is distinct to our on-the-fly approach.

**Generational Training and Distillation**. Stooke et al. (2021) recently introduced generational training, using policy distillation to transfer knowledge between generations, accelerating training. Our method is based on this idea, with changing generations. The use of distillation is further supported by Igl et al. (2021) who recently used this successfully to adapt to non-stationarities in reinforcement learning, however keeping hyperparameters and architectures fixed.

## 6 Conclusion & Discussion

In this paper, we propose BG-PBT: a new algorithm that significantly increases the capabilities of PBT methods for RL. Using recent advances in Bayesian Optimization, BG-PBT is capable of searching over drastically larger search spaces than previous methods. Furthermore, inspired by recent advances in generational learning, we show it is also possible to efficiently learn architectures

on the fly as part of a unified algorithm. The resulting method leads to significant performance gains across the entire Brax environment suite, achieving high performance even in previously untested environments. We believe BG-PBT is a significant step towards truly environment agnostic RL algorithms, while also offering a path towards open-ended learning where agents never stop tuning themselves and continuously expand their capabilities over time.

**Limitations & Future Work**. We note that while our method shows a significant boost in performance by including architectures for most environments, in some environments, such as Humanoid, we achieve better results without architecture search (Table 3). We hypothesize this is due to the complexity of the environment and an increased sensitivity to the network architecture. Furthermore, while we provide a theoretical guarantee for our method in Theorem 3.1 for searching purely over architectures, no such guarantees exist when we transfer between architectures across generations. Indeed, we occasionally see poor architectures being selected, which are then discarded during truncation selection. Therefore, an immediate future direction is to address these issues and to improve the architecture selection process. Another limitation is that while all RL-related hyperparameters are included in the search space, certain hyperparameters of BG-PBT could also be automatically searched for, including but not limited to distillation hyperparameters, which are currently fixed, and PBT parameters such as $t_{\text{ready}}$, which could allow us to avoid myopic and greedy behavior. Beyond these limitations, our algorithm readily transfers to other RL problems with high-dimensional mixed spaces, and thus we would readily accommodate more complicated architecture search spaces (e.g. vision-based environments) and incorporate environment parameters (Paul et al., 2016) into the search space to generalize to new tasks.

**Broader Impact**. We open-source our code so that practitioners in the field can accelerate their own deployment of RL systems. However, in doing so, we should be wary of the risk of also improving malicious use of RL; in particular, down-stream applications which could have an impact on people's security and privacy. To mitigate these risks, we encourage research on RL governance and safe RL. As a general purpose framework for improving any RL algorithm, our method should be part of that conversation.

**Acknowledgements**. XW and BR are supported by the Clarendon Scholarship at the University of Oxford. CL is funded by the Engineering and Physical Sciences Research Council (EPSRC). PB is funded through the Willowgrove Studentship. The authors would like to thank Yee Whye Teh for detailed feedback on an early draft and the anonymous AutoML conference reviewers & the area chair for their constructive comments which helped improve the paper.

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

# Supplementary Material

## A  Primer on GPs and BO

**Gaussian Processes.** In Bayesian Optimization (BO), Gaussian Processes, or GPs, act as *surrogate models* for a black-box function $f$ which takes an input $\mathbf{z}$ (in our case, the hyperparameters and/or the architecture parameters) and returns an output $y = f(\mathbf{z}) + \epsilon$ where $\epsilon \sim \mathcal{N}(0, \sigma^2)$. A GP defines a probability distribution over functions $f$ under the assumption that any finite subset $\{(\mathbf{z}_i, f(\mathbf{z}_i)\}$ follows a normal distribution (Rasmussen and Williams, 2006). Formally, a GP is denoted as $f(\mathbf{z}) \sim \text{GP}(m(\mathbf{z}), k(\mathbf{z}, \mathbf{z}'))$, where $m(\mathbf{z})$ and $k(\mathbf{z}, \mathbf{z}')$ are called the mean and covariance functions respectively, i.e. $m(\mathbf{z}) = \mathbb{E}[f(\mathbf{z})]$ and $k(\mathbf{z}, \mathbf{z}') = \mathbb{E}\left[(f(\mathbf{z}) - m(\mathbf{z}))(f(\mathbf{z}') - m(\mathbf{z}'))^T\right]$. The covariance function (kernel) $k(\mathbf{z}, \mathbf{z}')$ can be thought of as a similarity measure relating $f(\mathbf{z})$ and $f(\mathbf{z}')$. There have been various proposed kernels which encode different prior beliefs about the function $f(\mathbf{z})$ (Rasmussen and Williams, 2006).

If we assume a zero mean prior $m(\mathbf{z}) = 0$, to predict $f_* = f(\mathbf{z}_*)$ at a new data point $\mathbf{z}_*$, we have,

$$\begin{bmatrix} f \\ f_* \end{bmatrix} \sim \mathcal{N}\left(0, \begin{bmatrix} K & \mathbf{k}_*^T \\ \mathbf{k}_* & k_{**} \end{bmatrix}\right), \tag{3}$$

where $k_{**} = k(\mathbf{z}_*, \mathbf{z}_*)$, $\mathbf{k}_* = [k(\mathbf{z}_*, \mathbf{z}_i)]_{i \leq t}$, $t$ is the number of observed points for the GP, and $K = \left[k(\mathbf{z}_i, \mathbf{z}_j)\right]_{i,j \leq t}$. We denote our observations as $\{\mathbf{z}_1, f_1\}, \{\mathbf{z}_2, f_2\}, ..., \{\mathbf{z}_t, f_t\}$ and collect all past return observations as $\mathbf{f}_t = [f_1, ..., f_t]^\top$. Then, we may combine Eq. (3) with the fact that $p(f_* \mid f)$ follows a univariate Gaussian distribution $\mathcal{N}(\mu(\mathbf{z}_*), \sigma^2(\mathbf{z}_*))$. Given a new configuration $\mathbf{z}'$, the GP posterior mean and variance at $\mathbf{z}'$ may be computed as:

$$\mu_t(\mathbf{z}') := \mathbf{k}_t(\mathbf{z}')^T (\mathbf{K}_t + \sigma^2 \mathbf{I})^{-1} \mathbf{f}_t \tag{4}$$

$$\sigma_t^2(\mathbf{z}') := k(\mathbf{z}', \mathbf{z}') - \mathbf{k}_t(\mathbf{z}')^T (\mathbf{K}_t + \sigma^2 \mathbf{I})^{-1} \mathbf{k}_t(\mathbf{z}'), \tag{5}$$

where $\mathbf{K}_t := \{k(z_i, z_j)\}_{i,j=1}^t$ and $\mathbf{k}_t := \{k(z_i, z_t')\}_{i=1}^t$.

**Bayesian Optimization.** Bayesian optimization (BO) is a powerful sequential approach to find the global optimum of an expensive black-box function $f(\mathbf{z})$ without making use of derivatives. First, a surrogate model (in our case, a GP as discussed above) is learned from the current observed data $\mathcal{D}_t = \{\mathbf{z}_i, y_i\}_{i=1}^t$ to approximate the behavior of $f(\mathbf{z})$. Second, an *acquisition function* is derived from the surrogate model to select new data points that maximizes information about the global optimum – a common acquisition function that we use in our paper is the Upper Confidence Bound (UCB) (Srinivas et al., 2010) criterion which balances exploitation and exploration. Specifically, the UCB on a new, unobserved point $\mathbf{z}'$ is given by:

$$\text{UCB}(\mathbf{z}') = \mu_t(\mathbf{z}') + \sqrt{\beta_t} \sigma_t(\mathbf{z}'), \tag{6}$$

where $\mu_t$ and $\sigma_t$ are the posterior mean and standard deviation given in Eq. 4 above and $\beta_t > 0$ is a trade-off parameter between mean and variance. At each BO iteration, we find a batch of samples that sequentially maximizes the acquisition function above. The process is conducted iteratively until the evaluation budget is depleted, and the global optimum is estimated based on all the sampled data. In-depth discussions about BO beyond this brief overview can be found in recent surveys (Brochu et al., 2010; Frazier, 2018; Shahriari et al., 2016).

## B  Bayesian Optimization for PBT

In this section, we provide specific details for the modifications to CASMOPOLITAN to make it amenable for our setup which consists of non-stationary reward and a mixed, high-dimensional search space.

## B.1 Kernel Design

We use the following time-varying kernel ([Bogunovic et al., 2016](); [Parker-Holder et al., 2021]()) to measure the spatiotemporal distance between a pair of configuration vectors $\{\mathbf{z}, \mathbf{z}'\}$ with continuous, ordinal and/or categorical dimensions, and whose rewards are observed at timesteps $\{i, j\}$. For the most general case where we model all three types of variables, we have the following kernel function:

$$k(\mathbf{z}, \mathbf{z}', i, j) = \frac{1}{2}\Big(\big(k_x(\mathbf{x}, \mathbf{x}') + k_h(\mathbf{h}, \mathbf{h}')\big) + \big(k_x(\mathbf{x}, \mathbf{x}')k_h(\mathbf{h}, \mathbf{h}')\big)\Big)\Big((1-\omega)^{|i-j|/2}\Big) \tag{7}$$

where $\mathbf{x}$ denotes the continuous *and ordinal* dimensions and $\mathbf{h}$ denotes the categorical dimensions of the configuration vector $\mathbf{z}$, respectively, $k_x(\cdot, \cdot)$ is the kernel for continuous and ordinal inputs (by default Matérn 5/2), $k_{(h)}(\cdot, \cdot)$ is the kernel for the categorical dimensions (by default the exponentiated overlap kernel in [Wan et al. (2021)]()) and $\omega \in [0, 1]$ controls how quickly old data is decayed and is learned jointly during optimization of the GP log-likelihood. When the search space only contains continuous/ordinal variables, we simply have $k(\mathbf{z}, \mathbf{z}', i, j) = k_z(\mathbf{x}, \mathbf{x}')(1-\omega)^{|i-j|/2}$, and a similar simplification holds if the search space only contains categorical variables. We improve on [Parker-Holder et al. (2021)]() by directly supporting ordinal variables such as integers (for e.g. batch size) and selecting them alongside categorical variables using *interleaved acquisition optimization* as opposed to time-varying bandits which scales poorly to large discrete spaces.

## B.2 Proposing New Configurations

As discussed in App. [A](), a BO agent selects new configuration(s) by selecting those which maximize the acquisition function (in this case, the UCB acquisition function). This is typically achieved via off-the-shelf first-order optimizers, which is challenging in a mixed-input space as the discrete (ordinal and categorical) variables lack gradients and naïvely casting them into continuous variables yields invalid solutions which require rounding. To address this issue, [Parker-Holder et al. (2021)]() select $\mathbf{h}$ first via time-varying bandits (using the proposed TV.EXP3.M algorithm) and then select $\mathbf{x}$ by optimizing the BO acquisition function, *conditioned on* the chosen $\mathbf{h}$. This method scales poorly to spaces with a large number of categorical choices, as bandit problems generally require pulling each arm at least once. Instead, we develop upon *interleaved acquisition optimization* introduced in [Wan et al. (2021)]() which unifies all variables under a single GP, and alternates between optimization of the continuous and discrete variables:

---

**Algorithm 2** Interleaved optimization of acq($\mathbf{z}$)

---

1: **while** not converged **do**
2:     **Continuous**: Do a single step of gradient descent on the continuous dimensions.
3:     **Ordinal and Categorical**: Conditioned on the new continuous values, do a single step of local search: randomly select an ordinal/categorical variable and choose a different (categorical), or an adjacent (ordinal) value, if the new value leads to an improvement in acq($\cdot$).

---

Compared to the approach in [Wan et al. (2021)](), we include ordinal variables, which are optimized alongside the categorical variables via local search during acquisition optimization but are treated like continuous variables by the kernel. During acquisition, we define adjacent ordinals to be the neighboring values. For example, for an integer variable with a valid range $[1, 5]$ and current value 3, its neighboring values are 2 and 4. This allows us to exploit the natural ordering for ordinal variables whilst still ensuring that suggested configurations remain local and only explore *valid* neighboring solutions.

## B.3 Suggesting New Architectures

At the start of each generation for the full BG-PBT method, we have to suggest a pool of new architectures. For the first generation, we simply use random sampling across the joint space $\mathcal{J}$ to

fill up the initial population. For subsequent generations, we use a combination of BO and random sampling to both leverage information already gained from the architectures and allow sufficient exploration. For the BO, at the start of the $i$-th generation, we first fit a GP model solely in the architecture space $\mathcal{Y}$, by using the architectures from the $i-1$-th generation as the training data. Since these network architectures are trained with different hyperparameters during the generation, we use the *best return* achieved on each of these architectures as the training targets. We then run BO on this GP to obtain the suggestions for new architectures for the subsequent generation. In practice, to avoid occasional instability in the distillation process, we find it beneficial to select a number of architectures larger than $B$: we then start the distillation for all the agents, but use successive halving (Karnin et al., 2013) such that only $B$ agents survive and are distilled for the full budget allocated. By doing so, we trade a modest increase in training steps for greatly improved stability in distillation.

## B.4 Details on Trust Regions

To define trust regions for our time-varying objective, we again consider the most general case where the search space contains both categorical and continuous/ordinal dimensions. Given the configuration $\mathbf{z}_t^* = [\mathbf{h}_t^*, \mathbf{x}_t^*] = \arg\max_{\mathbf{z}_t}(f_t)$ with the best return at time $t$, we may define the trust region centered around $\mathbf{z}_t^*$:

$$
\text{TR}(\mathbf{z}_T^*) = \begin{cases} \left\{ \mathbf{h} \mid \frac{1}{d_h} \sum_{i=1}^{d_h} \delta(h_i, h_i^*) \le L_h \right\} & \text{for categorical } \mathbf{h}_T^* = \{h_i^*\}_{i=1}^{d_h} \\ \left\{ \mathbf{x} \mid |x_i - x_i^*| < \frac{\tilde{\ell}_i}{\prod_{i=1}^{d_x} \tilde{\ell}_i^{\frac{1}{d_x}}} L_x, 0 \le x_i \le 1 \right\} & \text{for continuous or ordinal } \mathbf{x}_T^* = \{x_i^*\}_{i=1}^{d_x}, \end{cases} \quad (8)
$$

where $\delta(\cdot, \cdot)$ is the Kronecker delta function, $L_h \in [0, 1]$ is the trust region radius defined in terms of normalized Hamming distance over the categorical variables, $L_x$ is the trust region radius defining a hyperrectangle over the continuous and ordinal variables, and $\{\tilde{\ell}_i = \frac{\ell_i}{\frac{1}{d_x} \sum_{i=1}^{d_x} \ell_i}\}_{i=1}^{d_x}$ are the normalized lengthscales $\{\ell_i\}$ learned by the GP surrogate over the continuous/ordinal dimensions. This means that the more sensitive hyperparameters, i.e. those with smaller learned lengthscales, will automatically be assigned smaller trust region radii.

For the restart of trust regions when either or both of the trust regions defined fall below some pre-defined threshold, we adapt the UCB-based criterion proposed in Wan et al. (2021) to the time-varying setting to re-initialize the population when a restart is triggered. For the $i$-th restart, we consider a global, auxiliary GP model trained on a subset of observed configurations and returns $D_{i-1}^* = \{\mathbf{z}_j^*, f_j^*\}_{j=1}^i$ and denote $\mu_g(\mathbf{z}; D_{i-1}^*)$ and $\sigma_g^2(\mathbf{z}; D_{i-1}^*)$ as the posterior mean and variance of the auxiliary GP. The new trust region center is given by the configuration $\mathbf{z}_i^{(0)}$ that maximizes the UCB score: $\mathbf{z}_i^{(0)} = \arg\max_{\mathbf{z} \in \mathcal{Z}} \mu_g(\mathbf{z}; D_{i-1}^*) + \sqrt{\beta_i} \sigma_g(\mathbf{z}; D_{i-1}^*)$ where $\beta_i$ is the UCB trade-off parameter. In the original CASMOPOLITAN, $D^*$ consists of the best configurations in all previous restarts $1, ..., i-1$, which is invalid for the time-varying setting. Instead, we construct $D_{i-1}^*$ using the following:

$$
D_{i-1}^* = \{\mathbf{z}_j^*, \mu_T(\mathbf{z}_j^*)\}_{j=1}^{i=1} \text{ where } \mathbf{z}_j^* = \arg\max_{\mathbf{z}_j \in \mathcal{D}_j} \mu_T(\mathbf{z}_j), \quad (9)
$$

where $\mathcal{D}_j$ denotes the set of previous configurations evaluated during the $j$-th restart and $\mu_T(\cdot)$ denotes the posterior mean of the time-varying GP surrogate *at the present timestep $t = T$*. Thus, instead of simply selecting the configurations of each restart that *led* to the highest observed return, we select the configurations that *would have led to the highest return if they were evaluated now*, according to the GP surrogate. Such a configuration preserves the convergence property of BG-PBT (without distillation and architecture search) shown in Theorem 3.1 and proven below in App. C.

## C Theoretical Guarantees

### C.1 Bound on the Maximum Information Gain

We start by deriving the maximum information gain, which extends the result presented in Wan et al. (2021) for the time-varying setting. Note that this result is defined over the number of local restarts $I$.

**Theorem C.1.** *Let* $\gamma(I; k; V) := \max_{A \subseteq V, |A| \le I} \frac{1}{2} \log |\mathbf{I} + \sigma^{-2} [k(\mathbf{v}, \mathbf{v}')]_{\mathbf{v},\mathbf{v}' \in A}|$ *be the maximum information gain achieved by sampling $I$ points in a GP defined over a set $V$ with a kernel $k$. Denote the constant* $\eta := \prod_{j=1}^{d_h} n_j$. *Then we have, for the time-varying mixed kernel $k$,*

$$\gamma(I; k; [\mathcal{H}, \mathcal{X}]) \lessgtr \frac{I}{\tilde{N}} \left( \lambda \eta \gamma(I; k_x; \mathcal{X}) + (\eta - 2\lambda) \log I + \sigma_f^{-2} \tilde{N}^3 \omega \right) \tag{10}$$

*where the time steps $\{1, ..., I\}$ are split into into $I/\tilde{N}$ blocks of length $\tilde{N}$, such that the function $f_t$ does not vary significantly within each block.*

*Proof.* Following the proof used in Bogunovic et al. (2016)), we split the time steps $\{1, ..., I\}$ into $I/\tilde{N}$ blocks of length $\tilde{N}$, such that within each block the function $f_i$ does not vary significantly. Then, we have that the maximum information gain of the time-varying kernel Bogunovic et al. (2016)) is bounded by

$$\gamma_I \le \left( \frac{I}{\tilde{N}} + 1 \right) \left( \tilde{\gamma}_{\tilde{N}} + \sigma_f^{-2} \tilde{N}^3 \omega \right)$$

where $\omega \in [0, 1]$ is the forgetting-remembering trade-off parameter, and we consider the kernel for time $1 - k_{time}(t, t') \le \omega |t - t'|$. We denote $\tilde{\gamma}_{\tilde{N}}$ as the maximum information gain for the time-invariant kernel counterpart in each block length of $\tilde{N}$.

Next, by using the bounds for the (time-invariant) mixed kernel in Wan et al. (2021) that $\tilde{\gamma}_{\tilde{N}} \le \mathcal{O}\big( (\lambda \eta + 1 - \lambda) \gamma(I; k_x; \mathcal{X}) + (\eta + 2 - 2\lambda) \log I \big)$, we get the new time-varying bound $\gamma(I; k; [\mathcal{H}, \mathcal{X}]) \lessgtr \frac{I}{\tilde{N}} \left( \lambda \eta \gamma(I; k_x; \mathcal{X}) + (\eta - 2\lambda) \log I + \sigma_f^{-2} \tilde{N}^3 \omega \right)$ where we have suppressed the constant term for simplicity. □

### C.2 Proof of Local Convergence in Each Trust Region

**Assumption C.2.** *The time-varying objective function $f_t(\mathbf{z})$ is bounded in $[\mathcal{H}, \mathcal{X}]$, i.e. $\exists F_l, F_u \in \mathbb{R}$ : $\forall \mathbf{z} \in [\mathcal{H}, \mathcal{X}], F_l \le f_t(\mathbf{z}) \le F_u, \forall t \in [1, ..., T]$.*

**Assumption C.3.** *Let us denote $L_{\min}^h$, $L_{\min}^x$ and $L_0^h, L_0^x$ be the minimum and initial TR lengths for the categorical and continuous variables, respectively. Let us also denote $\alpha_s$ as the shrinking rate of the TRs. The local GP approximates $f_t, \forall t \le T$ accurately within any TR with length $L^x \le \max \big( L_{\min}^x / \alpha_s, L_0^x (\lceil (L_{\min}^h + 1)/\alpha_s \rceil - 1)/L_0^h \big)$ and $L^h \le \max \big( \lceil (L_{\min}^h + 1)/\alpha_s \rceil - 1, \lceil L_0^h L_{\min}^x / (\alpha_s L_0^x) \rceil \big)$.*

**Theorem C.4.** *Given Assumptions C.2 & C.3, after a restart, BG-PBT converges to a local maxima after a finite number of iterations or converges to the global maximum.*

*Proof.* We may apply the same proof by contradiction used in Wan et al. (2021) for our time-varying setting, given the assumptions C.2 and C.3. For completeness, we summarize it below.

We show that our algorithm converges to (1) to a global maximum of $f$ (if does not terminate after a finite number of iterations) or (2) a local maxima of $f$ (if terminated after a finite number of iterations).

Case 1: when $t \to \infty$ and the TR lengths $L^h$ and $L^x$ have not shrunk below $L_{\min}^h$ and $L_{\min}^x$. From the algorithm description, the TR is shrunk after fail_tol consecutive failures. Thus, if

after $N_{\min} = \texttt{fail\_tol} \times m$ iterations where $m = \max(\lceil \log_{\alpha_e}(L_0^h/L_{\min}^h)\rceil, \lceil \log_{\alpha_e}(L_0^x/L_{\min}^x)\rceil)$, there is no success, BG-PBT terminates. This means, for case (1) to occur, BG-PBT needs to have at least one improvement per $N_{\min}$ iterations. Let consider the increasing series $\{f(\mathbf{z}^k)\}_{k=1}^{\infty}$ where $f(\mathbf{z}^k) = \max_{t=(k-1)N_{\min}+1,\dots,kN_{\min}}\{f(\mathbf{z}_t)\}$ and $f(\mathbf{z}_i)$ is the function value at iteration $t$. Thus, using the monotone convergence theorem (Bibby, 1974), this series converges to the global maximum of the objective function $f$ given that $f(\mathbf{z})$ is bounded (Assumption C.2).

Case 2: when BG-PBT terminates after a finite number of iterations, BG-PBT converges to a local maxima of $f(\mathbf{z})$ given Assumption C.3. Let us remind that BG-PBT terminates when either the continuous TR length $\leq L_{\min}^x$ or the categorical TR length $\leq L_{\min}^h$.

Let $L_s$ be the largest TR length that after being shrunk, the algorithm terminates, i.e., $\lfloor \alpha_s L_s \rfloor \leq L_{\min}^h$.[1] Due to $\lfloor \alpha_s L_s \rfloor \leq \alpha_s L_s < \lfloor \alpha_s L_s \rfloor + 1$, we have $L_s < (L_{\min}^h + 1)/\alpha_s$. Because $L_s$ is an integer, we finally have $L_s \leq \lceil (L_{\min}^h + 1)/\alpha_s \rceil - 1$. This means that $L_s = \lceil (L_{\min}^h + 1)/\alpha_s \rceil - 1$ is the largest TR length that after being shrunk, the algorithm terminates. We may apply a similar argument for the largest TR length (before terminating) for the continuous $L_{\min}^x/\alpha_s$.

In our mixed space setting, we have two separate trust regions for categorical and continuous variables. When one of the TR reaches its terminating threshold ($L_{\min}^x/\alpha_s$ or $\lceil (L_{\min}^h + 1)/\alpha_s \rceil - 1$), the length of the other one is $(\lceil L_0^h L_{\min}^x/(\alpha_s L_0^x) \rceil)$ or $L_0^x(\lceil (L_{\min}^h+1)/\alpha_s \rceil - 1)/L_0^h)$. Based on Assumption C.3, the GP can accurately fit a TR with continuous length $L^x \leq \max\left( L_{\min}^x/\alpha_s, L_0^x(\lceil (L_{\min}^h+1)/\alpha_s \rceil - 1)/L_0^h \right)$ and $L^h \leq \max\left( \lceil (L_{\min}^h + 1)/\alpha_s \rceil - 1, \lceil L_0^h L_{\min}^x/(\alpha_s L_0^x) \rceil \right)$. Thus, if the current TR center is not a local maxima, BG-PBT can find a new data point whose function value is larger than the function value of current TR center. This process occurs iteratively until a local maxima is reached, and BG-PBT terminates. $\qquad\square$

## C.3  Proof of Theorem 3.1

*Proof.* Under the time-varying setting, at the $i$-th restart, we first fit the global time-varying GP model on a subset of data $D_{i-1}^* = \{\mathbf{z}_j^*, f(\mathbf{z}_j^*)\}_{j=1}^{i-1}$, where $\mathbf{z}_j^*$ is the local maxima found after the $j$-th restart, or, a random data point, if the found local maxima after the $j$-th restart is same as in the previous restart.

Let $\mathbf{z}_i^{**} = \arg\max_{\forall \mathbf{z} \in [\mathcal{H}, \mathcal{X}]} f_t(\mathbf{z})$ [2] be the global optimum location at time step $i$. Let $\mu_{gl}(\mathbf{z}; D_{i-1}^*)$ and $\sigma_{gl}^2(\mathbf{z}; D_{i-1}^*)$ be the posterior mean and variance of the global GP learned from $D_{i-1}^*$. Then, at the $i$-th restart, we select the following location $\mathbf{z}_i^{(0)}$ as the initial centre of the new TR:

$$\mathbf{z}_i^{(0)} = \arg\max_{\mathbf{z} \in [\mathcal{H}, \mathcal{X}]} \mu_{gl}(\mathbf{z}; D_{i-1}^*) + \sqrt{\beta_i}\sigma_{gl}(\mathbf{z}; D_{i-1}^*),$$

where $\beta_i$ is the trade-off parameter in PB2 (Parker-Holder et al., 2020).

We follow Wan et al. (2021) to assume that at the $i$-th restart, there exists a function $g_i(\mathbf{z})$: (a) lies in the RKHS $\mathcal{G}_k([\mathcal{H}, \mathcal{X}])$ and $\|g_i\|_k^2 \leq B$, (b) shares the same global maximum $\mathbf{z}^*$ with $f$, and, (c) passes through all the local maxima of $f$ and any data point $\mathbf{z}'$ in $\mathcal{D}_{i-1}^* \cup \{\mathbf{z}_i^{(0)}\}$ which are not local maxima (i.e. $g_i(\mathbf{z}') = f(\mathbf{z}'), \forall \mathbf{z}' \in D_{i-1}^* \cup \{\mathbf{z}_i^{(0)}\}$). In other words, the function $g_i(\mathbf{z})$ is a function that passes through the maxima of $f$ whilst lying in the RKHS $\mathcal{G}_k([\mathcal{H}, \mathcal{X}])$ and satisfying $\|g_i\|_k^2 \leq B$.

Using $\beta_i$ defined in Theorem 2 in Srinivas et al. (2010) for function $g_i$, $\forall i$, $\forall z \in [\mathcal{H}, \mathcal{X}]$, we have,

$$\Pr\{|\mu_{gl}(\mathbf{z}; D_{i-1}^*) - g_i(\mathbf{z})| \leq \sqrt{\beta_i}\sigma_{gl}(\mathbf{z}; D_{i-1}^*)|\} \geq 1 - \zeta. \tag{11}$$

In particular, with probability $1 - \zeta$, we have that,

$$\mu_{gl}(\mathbf{z}_i^{(0)}; D_{i-1}^*) + \sqrt{\beta_i}\sigma_{gl}(\mathbf{z}_i^{(0)}; D_{i-1}^*) \geq \mu_{gl}(\mathbf{z}_i^{**}; D_{i-1}^*) + \sqrt{\beta_i}\sigma_{gl}(\mathbf{z}_i^{**}; D_{i-1}^*) \geq g_i(\mathbf{z}_i^{**}). \tag{12}$$

---

[1] The operator $\lfloor . \rfloor$ denotes the floor function

[2] Notationally, at the $i$-th restart, $\mathbf{z}_i^{**}$ is the global optimum location while $\mathbf{z}_i^*$ is the local maxima found by BG-PBT.

Thus, $\forall i$, with probability $1 - \zeta$ we have

$$g_i(\mathbf{z}_i^{**}) - g_i(\mathbf{z}_i^{(0)}) \leq \mu_{gl}(\mathbf{z}_i^{(0)}; D_{i-1}^*) + \sqrt{\beta_i}\sigma_{gl}(\mathbf{z}_i^{(0)}; D_{i-1}^*) - g_i(\mathbf{z}_i^{(0)}) \leq 2\sqrt{\beta_i}\sigma_{gl}(\mathbf{z}_i^{(0)}; D_{i-1}^*).$$

Since $g_i(\mathbf{z}_i^{(0)}) = f(\mathbf{z}_i^{(0)})$, and $g_i(\mathbf{z}_i^{**}) = f(\mathbf{z}_i^{**})$, hence, $f_i(\mathbf{z}_i^{**}) - f(\mathbf{z}_i^{(0)}) \leq 2\sqrt{\beta_i}\sigma_{gl}(\mathbf{z}_i^{(0)}; D_{i-1}^*)$ with probability $1 - \zeta$. With $\mathbf{z}_i^*$ as the local maxima found by BG-PBT at the $i$-th restart. As $f(\mathbf{z}_i^{(0)}) \leq f(\mathbf{z}_i^*)$, we have,

$$f_i(\mathbf{z}_i^{**}) - f_i(\mathbf{z}_i^*) \leq 2\sqrt{\beta_i}\sigma_{gl}(\mathbf{z}_i^{(0)}; D_{i-1}^*). \tag{13}$$

Let $\mathbf{z}_{i,b}$ be the point chosen by our algorithm at iteration $i$ and batch element $b$, we follow Parker-Holder et al. (2020) to define the time-varying instantaneous regret as $r_{i,b} = f_i(\mathbf{z}_i^{**}) - f_i(\mathbf{z}_{i,b})$. Then, the time-varying batch instantaneous regret over $B$ points is as follows

$$r_i^B = \min_{b \leq B} r_{i,b} = \min_{b \leq B} f_i(\mathbf{z}_i^{**}) - f_i(\mathbf{z}_{i,b}), \forall b \leq B \tag{14}$$

Using Equation (13) and Theorem 2 in Parker-Holder et al. (2020), we bound the cumulative batch regret over $I$ restarts and $B$ parallel agents

$$R_{IB} = \sum_{i=1}^{I} r_i^B \leq \sqrt{\frac{C_1 I \beta_I}{B} \gamma(IB; k; [\mathcal{H}, \mathcal{X}])} + 2 \tag{15}$$

where $C_1 = 32/\log(1 + \sigma_f^2)$, $\beta_I$ is the explore-exploit hyperparameter defined in Theorem 2 in Parker-Holder et al. (2020) and $\gamma(IB; k; [\mathcal{H}, \mathcal{X}]) \lesssim \frac{IB}{\tilde{N}}\left(\lambda\eta\gamma(I; k_x; \mathcal{X}) + (\eta - 2\lambda)\log IB + \sigma_f^{-2}\tilde{N}^3\omega\right)$ is the maximum information gain defined over the mixed space of categorical and continuous $[\mathcal{H}, \mathcal{X}]$ in the time-varying setting defined in Theorem C.1.

$\square$

We note that given Theorem 3.1, if we use the squared exponential kernel over the continuous variables, $\gamma(\tilde{N}B; k; \mathcal{X}) = \mathcal{O}([\log \tilde{N}B]^{d+1})$ (Srinivas et al., 2010), the bound becomes $R_{IB} \leq \sqrt{\frac{C_1 I^2 \beta_I}{\tilde{N}}\left(\lambda\eta\left[\log \tilde{N}B\right]^{d+1} + (\eta - 2\lambda)\log IB + \sigma_f^{-2}\tilde{N}^3\omega\right)} + 2$ where $\tilde{N} \leq I$, $B \ll T$ and $\omega \in [0, 1]$.

# D  Full PPO Hyperparameter Search Space

We list the full search space for PPO in Table 4. The architecture and hyperparameters form the full 15-dimensional mixed search space. For methods that do not search in the architecture space (e.g., PBT, PB2, random search baselines in $\mathcal{Z}$, and the partial BG-PBT in Ablation Studies that uses §3.1 only), the last 6 dimensions are fixed to the default architecture used in BRAX: a policy network with 4 hidden layers each containing 32 neurons, and a value network with 5 hidden layers each containing 256 neurons. By default, spectral normalization is disabled in both networks.

# E  Implementation Details

We list the hyperparameters for our method BG-PBT in Table 5. Since BG-PBT uses the CASMOPOLITAN BO agent, it also inherits the default hyperparameters from Wan et al. (2021) which are used in all our experiments (Table 6). We refer the readers to App. B.5 of Wan et al. (2021) which examines the sensitivity of these hyperparameters. Note that in our current instantiation, we use $\alpha_V = 0$ so we only transfer policy networks across generations, since we found the value function was

Table 4: The hyperparameters for PPO form a 15-dimensional mixed search space.

| Hyperparameter | Type | Range |
|---|---|---|
| learning rate | log-uniform | [1e-4, 1e-3] |
| discount factor ($\gamma$) | uniform | [0.9, 0.9999] |
| entropy coefficient (c) | log-uniform | [1e-6, 1e-1] |
| unroll length | integer | [5, 15] |
| reward scaling | uniform | [0.05, 20] |
| batch size | integer (power of 2) | [32, 1024] |
| no. updates per epoch | integer | [2, 16] |
| GAE parameter ($\lambda$) | uniform | [0.9, 1] |
| clipping parameter ($\epsilon$) | uniform | [0.1, 0.4] |
| $\pi$ network width | integer (power of 2) | [32, 256] |
| $\pi$ network depth | integer | [1, 5] |
| $\pi$ use spectral norm | binary | [True, False] |
| $V$ network width | integer (power of 2) | [32, 256] |
| $V$ network depth | integer | [1, 5] |
| $V$ use spectral norm | binary | [True, False] |

Table 5: Hyperparameters for BG-PBT.

| Hyperparameter | Value | Description |
|---|---|---|
| $B$ | 8 | Population size (number of parallel agents) |
| $q$ | 12.5 | % agents replaced each iteration ($q$) |
| $t_{\max}$ | 150M | Total timesteps |
| $\alpha_{\text{RL}}$ | 1 | RL weight |
| $\alpha_V$ | 0 | Value function weight |
| $\alpha_{\pi}$ | 5 | Policy weight |

less informative. We linearly anneal the coefficients for the supervised loss $\alpha_V$ and $\alpha_{\pi}$ from their original value to 0 over the course of the distillation phase. This means we smoothly transition to a pure RL loss over the initial part of each new generation.

Our method is built using the PyTorch version of the BRAX (Freeman et al., 2021) codebase at https://github.com/google/brax/tree/main/brax. The codebase is open-sourced under the Apache 2.0 License. The BRAX environments are often subject to change, for full transparency, our evaluation is performed using the 0.10.0 version of the codebase. We ran all our experiments on Nvidia Tesla V100 GPUs and used a single GPU for all experiments. We note that the PPO baseline used in Table 1 is implemented in a different framework (JAX) to ours, which has some differences in network weight initialization. The hyperparameters for the PPO baseline are tuned via grid-search on a reduced hyperparameter search space (Freeman et al., 2021). Since no hyperparameters were provided for the Hopper environment, we use the default in Freeman et al. (2021).

For all experiments, we use $T_{\max} = 150M$, population size (number of parallel agents) $B = 8$ and $q = 12.5$ (percentage of the agents that are replaced at each PBT iteration – in this case, at each iteration, the single worst-performing agent is replaced). For all environments except for Humanoid and Hopper, we use a fixed $t_{\text{ready}} = 1M$. To avoid excessive sensitivity to initialization, at the beginning of training for all PBT-based methods (PB2, PBT and BG-PBT) we initialize with 24 agents and train for $t_{\text{ready}}$ steps and choose the top-$B$ agents as the initializing population. For the full BG-PBT, to trigger distillations and hence a new generation, we set a patience of 20 (i.e., if the evaluated return fails to improve after 20 consecutive $t_{\text{ready}}$ steps, a new generation is started). Since starting new generations can be desirable even if the training has not stalled, we introduce a second

Table 6: Hyperparameters for BG-PBT inherited from CASMOPOLITAN

| Hyperparameter | Value | Description |
|---|---|---|
| TR multiplier | 1.5 | multiplicative factor for each expansion/shrinking of the TR. |
| succ_tol | 3 | number of consecutive successes before expanding the TR |
| fail_tol | 10 | number of consecutive failures before shrinking the TR |
| Min. continuous TR radius | 0.15 | min. TR of the continuous/ordinal variables before restarting |
| Min. categorical TR radius | 0.1 | min. TR of the categorical variables before restarting |
| Init. continuous TR radius | 0.4 | initial TR of the continuous/ordinal variables |
| Init. categorical TR radius | 1 | initial TR of the categorical variables |

criterion to also start a new generation after 40M steps. Thus, a new generation is started when either criterion is met (40M steps since last distillation, or 20 consecutive failures in improving the evaluated return). For distillation at the start of every generation (all except initial), we begin distillation with 24 agents (4 suggested by BO and the rest from random sampling, see App. B.3 for details). We then use successive halving to only distill $B$ of them using the full budget of 30M steps with the rest terminated early.

For the Humanoid and Hopper environments, we observed that PBT-style methods performed poorly across the board (See App. F for detailed results): in particular, on Hopper we notice that agents often learn a sub-optimal mode where it only learns to stand up (hence collecting the reward associating with simply surviving) but not to move. On Humanoid, we find that agents often learn a mode where the humanoid does not use its knee joint – in both cases, the agents seem to get stuck in stable but sub-optimal modes which use fewer degrees-of-freedom than they are capable of exploiting. This behavior was ameliorated by linearly annealing the interval $t_{\text{ready}}$ from 5M to 1M as a function of timesteps to not encourage myopic behavior at the start. Since the increase in $t_{\text{ready}}$ at the initial stage of training will lead to more exploratory behaviors, we increase the threshold before triggering a new generation at 60M for these two environments.

## F  Additional Experiments

### F.1  Scalability with Increased Training Budget and/or $B$

For more complicated large-scale environments, PBT is often used with a much larger population size than what we present in this paper ($B = 8$). In this section, we investigate whether BG-PBT benefits from increased parallelism by increasing the number of agents to $B = 24$ and training for much longer. In this instance, we use the Hopper environment as a testbed as it is amongst the most challenging in the current version of BRAX and is particularly well suited to PBT-style methods. We show the results in Fig. 5: for a $t_{\text{max}}$ of 150M, there is a small improvement over the $B = 8$ results presented in the main text; whereas there is a significant benefit from jointly scaling up $B$, $t_{\text{max}}$ and $t_{\text{ready}}$ as exemplified by the 300M_10M result.

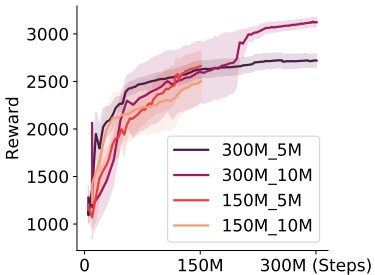

Figure 5: Evaluation of BG-PBT on the Hopper environment with $B = 24$, and varying $t_{\text{max}}$ and $t_{\text{ready}}$. Results in the format "{$t_{\text{max}}$}_{$t_{\text{ready}}$}".

### F.2  Effects of Reduced Training Budget

We additionally conduct experiments on BG-PBT with the maximum timesteps roughly halved from the default 150M used in the main experiments to remove the effect of the additional samples used during distillation. We show the results in Fig. 6, where BG-PBT_s denotes BG-PBT run for roughly 75M steps. Compared to the training setup outlined in App. E, to further reduce the training cost, we also reduce the number of initializing population to 12, reduce the distillation timesteps to 20M and allow for only one generation of distillation. The results show that BG-PBT still performs well with results on par with or exceeding previous baselines using the full budget.

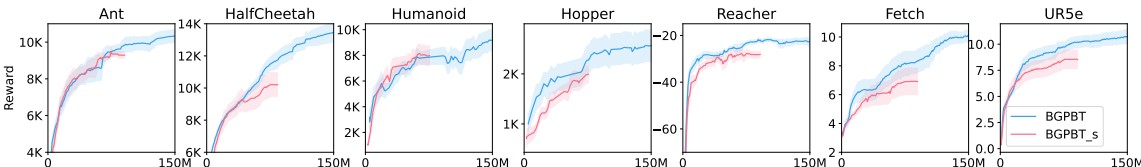

Figure 6: Mean evaluated return over the population with ±1 SEM (shaded) across 7 random seeds in all environments.

## F.3 Components of BG-PBT

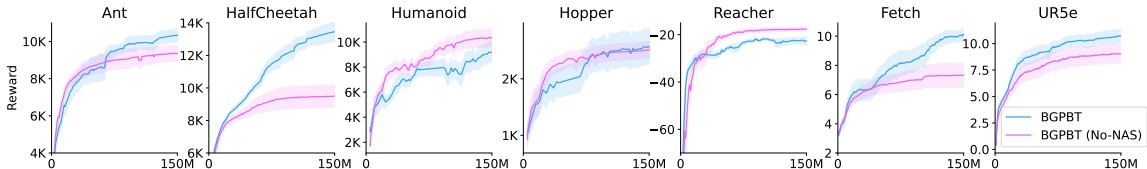

Figure 7: Comparison of full BG-PBT and BG-PBT without NAS and distillation. Mean evaluated return over the population shown with ±1 SEM (shaded) across 7 random seeds in all environments.

We show the training trajectories of BG-PBT and the variant of BG-PBT without neural architecture search and distillation in Fig. 7. We also perform additional ablation studies on components of BG-PBT on Ant and HalfCheetah in Fig. 8. The modifications to BG-PBT we consider are:

1. No NAS/TR PBT without TR-based BO or NAS. This is identical to the PB2 baseline described in the main text.

2. No NAS BG-PBT with TR-based BO in Sec. 3.1 but without NAS or distillation.

3. Random Arch BG-PBT with TR-based BO and distillation, but at the start of each generation, the architectures are selected randomly instead using BO+RS followed by the successive halving strategy described in App. B.3.

4. Static Arch BG-PBT with TR-based BO and distillation, but without NAS: all agents are started with the same default architectures for both the policy and value networks, and at the start of a new generation we distill across identical architectures.

The results further demonstrate the benefit of introducing both TR-based BO and NAS to PBT-style methods in BG-PBT. The results in Fig. 8 also highlight the importance of having architecture diversity for distillation to be successful – for both environments, removing the architecture variability (Static Arch) led to a significant drop in performance, which in some cases even under-performed the baseline without any distillation. In contrast, simply initializing the new agents with random architectures performed surprisingly well (Random Arch). This provides more even evidence that optimal architectures at different stages of training may vary, and thus should also vary dynamically through time.

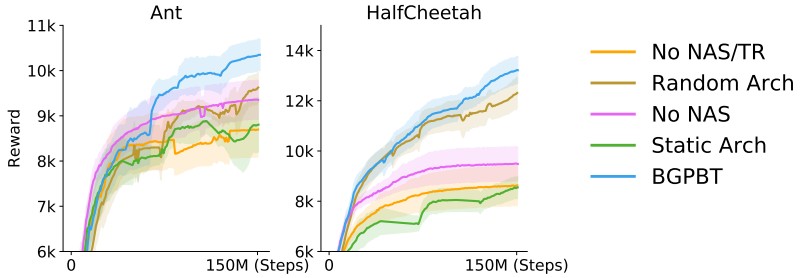

Figure 8: Additional ablation studies on the Ant and HalfCheetah environments.

## F.4 Hyperparameter and Architecture Schedules Learned on Additional Environments

Supplementary to Fig. 4, we show the hyperparameter and architectures schedules learned by BG-PBT in Fig. 9 for additional environments where architecture search improves performance. We see similar trends to the schedules learned for the Ant environment.

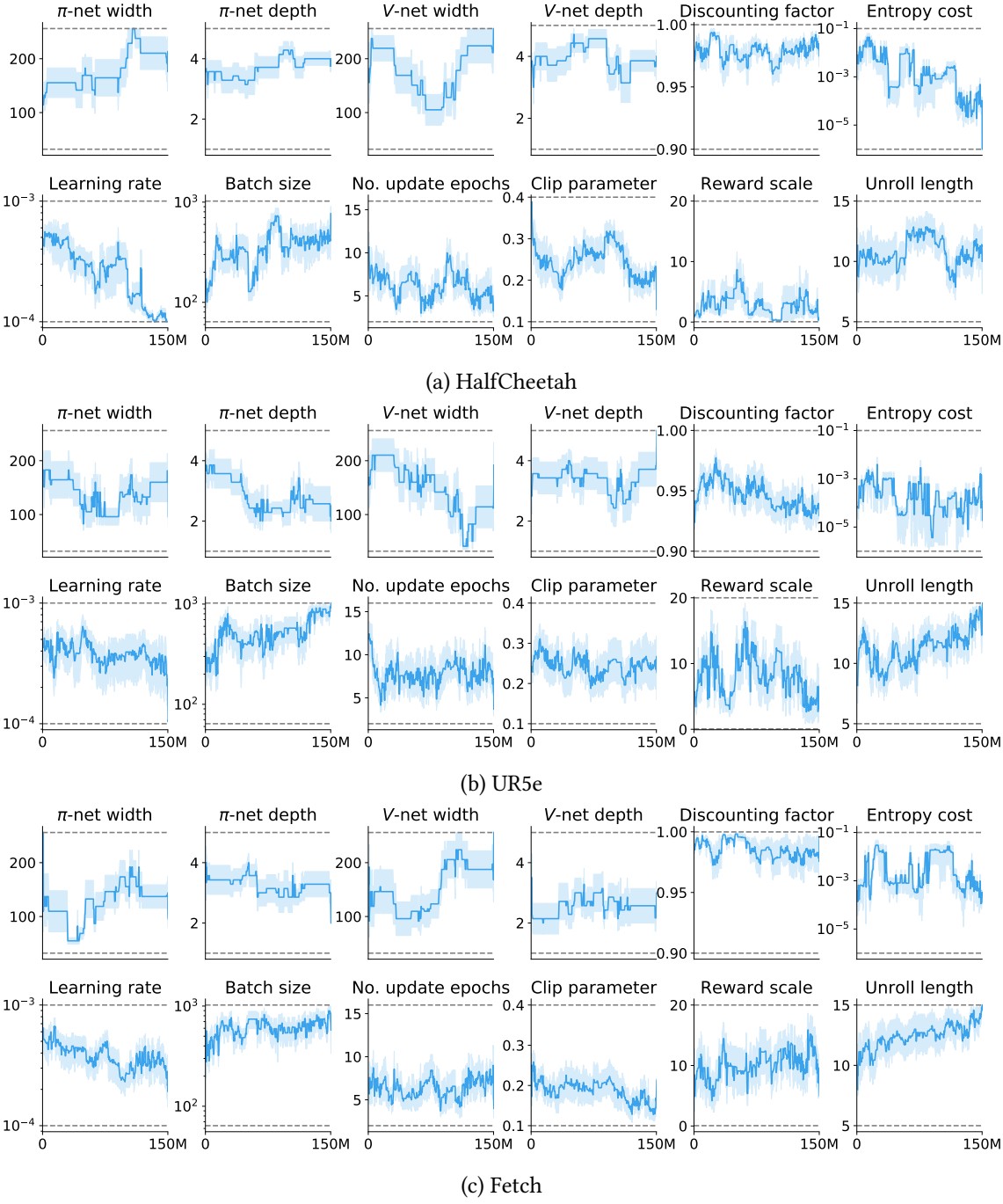

(a) HalfCheetah

(b) UR5e

(c) Fetch

Figure 9: The hyperparameter and architecture schedule discovered by BG-PBT on various environments: we plot the hyperparameters of the best-performing agent in the population averaged across 7 seeds with ± 1 SEM shaded. Gray dashed lines denote the hyperparameter bounds.

### F.5 Hopper and Humanoid Results with Constant $t_{\mathrm{ready}} = 1M$

For our main evaluation, we linearly anneal $t_{\mathrm{ready}}$ through time. We show why this is necessary in Table 7 by evaluating the PBT-style methods with a constant $t_{\mathrm{ready}} = 1M$ as with the other environments. We observe a considerable decrease in evaluated return compared to the results in Table 1.

Table 7: Mean evaluated return ±1sem across 7 seeds shown for Humanoid and Hopper environments with constant $t_{\mathrm{ready}} = 1M$.

| Method
Search space | PBT
$\mathcal{Z}$ | PB2
$\mathcal{Z}$ | BG-PBT
$\mathcal{J}$ |
|---|---|---|---|
| Humanoid | $7498_{\pm 666}$ | $7667_{\pm 1000}$ | $7949_{\pm 876}$ |
| Hopper | $1667_{\pm 222}$ | $1253_{\pm 77}$ | $2257_{\pm 290}$ |

### F.6 Ablations on Explicit Treatment of Ordinal Variables

For bg-pbt, we extended Casmopolitan (Wan et al., 2021) by further accommodating for ordinal variables (i.e., discrete variables with ordering, such as integer variables for the width of a multi-layer perceptron). In this section, we show the empirical benefits of this over treating them as categorical variables (the default in Wan et al. (2021)).

We consider 3 Brax environments (Fig. 10), where we further compare the full bg-pbt with a variant of bg-pbt where we use the original Casmopolitan by treating the ordinal variables ($\log_2$-batch size, widths and depths of the value and policy MLPs, unroll length, number of updates per epoch) as categorical variables. It is clear that the ordinal treatment in our case results in better performance in all 3 environments.

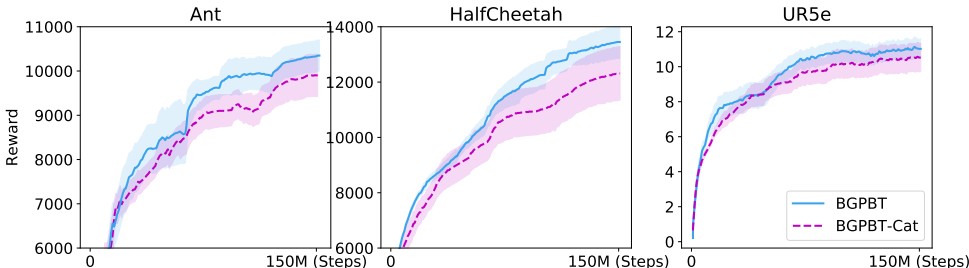

Figure 10: Comparison of bg-pbt with the variant of bg-pbt that uses original Casmopolitan (bg-pbt-Cat) that treats ordinal variables as categorical.

### F.7 Comparison Against Random Search with Manually Defined Learning Rate Schedule

We compare against random search that uses an equivalent amount of compute resources, but with a manually defined learning rate schedule, in Table 8. Specifically, we use random search for all hyperparameters in either the joint search space ($\mathcal{J}$) or the hyperparameter search space ($\mathcal{Z}$) defined in Table 1, with the exception that instead of using flat learning rates, we search for an *initial* learning rate which is cosine annealed to $10^{-8}$ by the end of the training. We find that in Ant and HalfCheetah where the bg-pbt discovered schedules are similar to the manual cosine schedules, RS-Anneal significantly outperforms regular RS, but when the discovered schedules deviate from the manual design in the case of UR5e, we find the margin of improvement to be much smaller. This shows that there is unlikely to be an optimal manual schedule for all environments, further demonstrating the desirable flexibility of bg-pbt in adapting to different environments.

Table 8: Comparison against RS and RS with cosine annealing (RS-Anneal). The results for RS and BG-PBT are lifted from Table 1.

| Method Search space | RS $\mathcal{Z}$ | RS $\mathcal{J}$ | RS-Anneal $\mathcal{Z}$ | RS-Anneal $\mathcal{J}$ | BG-PBT $\mathcal{J}$ |
|---|---|---|---|---|---|
| Ant | $6780_{\pm 317}$ | $4781_{\pm 515}$ | $9640_{\pm 79}$ | $9536_{\pm 257}$ | $\mathbf{10349_{\pm 326}}$ |
| HalfCheetah | $9502_{\pm 76}$ | $10340_{\pm 329}$ | $9672_{\pm 157}$ | $\mathbf{13071_{\pm 360}}$ | $\mathbf{13450_{\pm 551}}$ |
| UR5e | $5.3_{\pm 0.4}$ | $6.9_{\pm 0.4}$ | $7.7_{\pm 0.3}$ | $7.8_{\pm 0.4}$ | $\mathbf{10.7_{\pm 0.6}}$ |

Furthermore, in all environments, BG-PBT still outperforms RS, with or without manual learning rate scheduling. This is particularly notable as previous PBT-style methods were often known to under-perform random search, especially with small population sizes.

