# OpenReview forum: "Bayesian Generational Population-Based Training"
_automl.cc/AutoML/2022/Track/Main — AutoML-Conf 2022 (Main Track)_

### Official Review · Reviewer_WUDH · 2022-03-31

**Potential Impact On The Field Of Automl Rating:** 2
**Technical Quality And Correctness Rating:** 2
**Clarity Rating:** 3

**Summary Of Contributions:**

The author extends Casmopolitan (Wan et al., 2021) and PB2-Mix (Parker-Holder et al., 2021) by introducing the BG-PBT algorithm to jointly optimize hyperparameters and architecture of the value/policy network. The novelty lies in the explicit consideration of ordinal variables and the architecture search. They empirically evaluate their approach with the use of PPO in continuous control tasks of the BRAX library. They claim to provide the first approach for architecture search in population-based frameworks and that their approach creates an automatic architecture curriculum to improve significantly higher performance.


**Clarity:**

The paper is well-written and understandable. Sometimes the authors cite a bit misleading related work, e.g. “PPO approximates TRPO (Schulman et al., 2015)” does only describe TRPO but not that PPO approximates TRPO or “a time-varying Gaussian Process (GP) (Rasmussen and Williams, 2006)” is the cite only for GP but the correct cite for a time-varying GP could be Bogunovic et. al 2016. Also, the Appendix contains author comments in red.


**Overall Review:**

The paper considers the important area of AutoRL in combination with architecture search and argues that the architecture should be part of the HPO. The authors who the benefits of their approach in experiments on Brax where they outperformed other AutoRL approaches.

My main criticism is that the paper is not the first work in AutoRL including neural architecture search into population-based optimization. The work “Sample-Efficient Automated Deep Reinforcement Learning” at ICLR 2021 as well as “Neural Architecture Evolution in Deep Reinforcement Learning for Continuous Control” at the Workshop on Meta-Learning, NeurIPS 2019, already jointly optimized hyperparameter and architecture parameter in a population-based fashion similar to PBT but focused on off-policy instead of on-policy RL. Some findings, like the decrease in the learning rate, are similar to the work above. It also does not consider this work as related work.
Furthermore, it is unclear if it is beneficial to treat ordinal hyperparameters like the batch size as ordinal instead of as categorical parameters in a BO setting. How much does the optimization benefit from this? And how would a random search perform if the search space would include a learning rate schedule instead of keeping it fixed?


**Potential Impact On The Field Of Automl:**

AutoRL which includes the architecture of an agent in the search space is an important topic in the field of AutoML as well as for applied RL. However, I doubt that this work is the first which considers network architecture search and I am also skeptical about the validity of the experiments. Therefore, I would expect a low impact on the field of AutoML.


**Reproducibility:**

The reproducibility list is completely filled out and there is source code available but the code itself is not well documented and contains todos. I did not test if the code is running.


**Review Confidence:**

5: You are absolutely certain about your assessment. You are very familiar with the related work and checked all the details carefully.

**Review Rating:**

2: Reject, not good enough

**Review Summary:**

I suppose the authors are not aware of the related work mentioned above. However, since it exists, the paper lacks novelty due to existing work on joint hyperparameters and network parameter optimization in a population-based fashion. Additionally, I have doubts about the experiments and the benefit of considering some parameters as ordinal scales.

**Technical Quality And Correctness:**

The proposed approach seems feasible. It was hard for me to follow the proof of the local convergence. The experiments seem plausible. However, the ablation studies in  Table 3 seem confusing, since they compare an AutoRL approach which finds the optimal architecture with a “default BRAX architecture” (No NAS) which was found with unknown amounts of computing. Or, the conclusion could be that BG-PTB is in 3 of 7 environments not able to find the optimal (No NAS) or optimal performing architecture. I also was wondering why the authors do not consider PB2-Mix as related work in the experiments.

---

### Official Review · Reviewer_N2A6 · 2022-04-07

**Potential Impact On The Field Of Automl:** N/A for reproducibility reviewers
**Potential Impact On The Field Of Automl Rating:** 4
**Technical Quality And Correctness:** N/A for reproducibility reviewers
**Technical Quality And Correctness Rating:** 4
**Clarity:** N/A for reproducibility reviewers
**Clarity Rating:** 4

**Summary Of Contributions:**

N/A for reproducibility reviewers

**Ethics Details (Optional):**

N/A for reproducibility reviewers

**Overall Review:**

N/A for reproducibility reviewers

**Reproducibility:**

The authors present several experiments to prove their statements. The claims related to the reproducibility part of the checklist are coherent. The work is provided with the code. The code is structured and has a good overall quality. The code is provided for all the experiments presented in the paper, as well as the command needed to launch it.
I tried to test the code on my device without a cuda-based graphics card. During the installation process, there are issues provided by the python modules which require cuda. I tried to remove it, substituting the modules with their non-cuda version but the code still did not run. I see in the code that there are checks to switch between the cuda e non-cuda versions of the routines, the problems are in the installation phase. I suggest to create two new requirements files, one for cuda-based devices and one for devices without cuda. I also suggest to include only the required packages on top of the basic python installation and avoid to provide all packages, in order to avoid conflicts due to useless packages.
Overall, I think there are no particular issues to mention about the code on devices with cuda.

The numerical values inserted in the multiple-choice fields must not be taken into account.

**Review Confidence:**

5: You are absolutely certain about your assessment. You are very familiar with the related work and checked all the details carefully.

**Review Rating:**

6: Strong accept, should be highlighted

**Review Summary:**

N/A for reproducibility reviewers

---

### Official Review · Reviewer_98ZS · 2022-04-08

**Potential Impact On The Field Of Automl Rating:** 2
**Technical Quality And Correctness Rating:** 3
**Clarity Rating:** 3

**Summary Of Contributions:**

This paper proposes an approach that seeks to augment existing Population-based Training  (PBT) methods by a) improving hyperparameter search by incorporating recent advances in Bayesian Optimization (BO) for high-dimensional mixed-input search spaces, and b) improving neural architecture search (NAS) using generational training with on-policy distillation. This results in a unified approach for simultaneous hyperparameter and neural architecture search.

**Clarity:**

The writing in this manuscript is of high quality, with very few spelling and grammatical errors. It is generally well-structured, concise, and quite clear. The preliminaries provided make the manuscript mostly self-contained, which is a welcome quality.

**Overall Review:**

Overall, the paper is well-written and the proposed method is clearly communicated. The method has some novelties (combines in a nontrivial manner things that work well in isolation) which are empirically validated fairly extensively.  The methodologies are highly relevant to the AutoML community. However, it is unclear whether they will have a significant and lasting impact given the lack of principled grounding.

**Potential Impact On The Field Of Automl:**

The improvements to PBT methods as summarized above are likely to have a small to medium positive impact in the area of AutoML, particularly given that the accompanying source code has been made available.  Other concerns relating to impact are discussed under subsequent sections.

**Reproducibility:**

The minute details necessary to reproduce the experimental results appear quite overwhelming from reading the paper alone. Though the accompanying code has been provided, which greatly helps, there are still some minor details that have been obscured, and which I could not find either in the appendix or from browsing the source code. An example of this is on pg. 6 line 201 "we use suggestions from BO and/or random search with successive halving over the architecture space". This seems to be a tiny component in the grander scheme and the contributions resulting from their hyperparameter settings could just be minor, but all of these tiny components combined together tend to have a large compounding effect that ultimately makes it impossible to reproduce the results from the proposed method. There appear to be many small pieces like this in the proposed method that come without a sufficiently detailed description. Therefore, I am not very confident that I could reproduce the results.

**Review Confidence:**

4: You are confident in your assessment, but not absolutely certain. It is unlikely, but not impossible, that you did not understand some parts of the submission or that you are unfamiliar with some pieces of related work.

**Review Rating:**

4: Marginally above the acceptance threshold (use sparsely)

**Review Summary:**

To summarize, I believe the positives outweigh the negatives in this contribution, and I would therefore weakly recommend acceptance of this paper. I believe it is highly relevant and will be of great interest to the AutoML community.

**Technical Quality And Correctness:**

Each of the individual components (i.e., trust-region BO, policy distillation, etc.) is technically sound and, by extension, so is their combination in the proposed PBT framework.  One flaw is that the proposed method becomes somewhat convoluted, with many of its own hyperparameters (q, t_ready, etc.) Furthermore, the design choices are also perhaps also slightly arbitrary in that it simply combines components that work well in isolation into a unified whole in such a way that it works well for the problems being considered. There appears to be no real underlying theory or principle that drives these design choices.

Case in point: the fact that the authors have to linearly anneal t_ready from 5M to 1M for some problems and not others, with no justification other reason than this yielding better results.

Regarding the quality of the experimental design, one obvious concern is that the method may have been overfitted to the problem of tuning the PPO algorithm on the Brax test suite. This may or may not be a significant concern, given that different tasks in the same suite have different behaviours and dynamics, which appears to be the case.

---

### Official Review · Reviewer_xj4C · 2022-04-10

**Potential Impact On The Field Of Automl Rating:** 4
**Technical Quality And Correctness Rating:** 4
**Clarity Rating:** 4

**Summary Of Contributions:**

This paper introduces BG-PBT, a novel algorithm for self-tuning in reinforcement learning. This algorithm improves PBT methods by adding architecture search using distillation in addition to hyperparameters tuning.

**Clarity:**

The paper is overall very clear, notably on central concepts of BG-BPT such as population-based training, trust regions or network distillation.

**Overall Review:**

This is a very good paper that introduces a novel self-tuning algorithm and reaches state-of-the-art performance in Brax environments. It is well written with an extensive literature review. The algorithm is rigorously benchmarked and analysed.

**Potential Impact On The Field Of Automl:**

BG-PBT achieves new state-of-the-art performance on all seven Brax environments used for the experiments. This demonstrates the strength of the new ideas from this paper and has a very high potential impact on the field of AutoML.

**Reproducibility:**

The experiments are reproducible with a clear README in the code.

**Review Confidence:**

4: You are confident in your assessment, but not absolutely certain. It is unlikely, but not impossible, that you did not understand some parts of the submission or that you are unfamiliar with some pieces of related work.

**Review Rating:**

6: Strong accept, should be highlighted

**Review Summary:**

Given the quality of the paper and the performance of the algorithm, this paper should be highlighted.

**Technical Quality And Correctness:**

The technical quality of the paper is very good. The environments used for experiments are pretty standard and relevant to benchmark BG-PBT. The literature review is very extensive and BG-PBT is benchmarked against relevant algorithms. The authors provide an interesting analysis of all methods with and without architecture search (when relevant) as well as the evolution of hyperparameters during training.

---

### Meta-Review · Area_Chair_W1H7 · 2022-05-08

**Recommendation:** Accept
**Confidence:** 4

**Metareview:**

I would like to thank all reviewers for their constructive comments, which have improved the quality of the submission!

In my view, the authors are able to fix the issues raised by WUDH & 98ZS; the mentioned related works do not settle several outstanding questions in AutoRL; some of these are well addressed by the present submission, as pointed out by reviewers xj4C & 98ZS, even if novelty claims need to be reduced.

While WUDH has perhaps most contributed to improving the quality of the manuscript, I believe they were successful in doing so, and hence I cannot support their original evaluation that the relevance of this work to AutoRL is small.

Since the crucial concerns seem to have been addressed to a satisfactory degree, I hence recommend acceptance.

---

### Decision · Program_Chairs · 2022-05-13

Accept